# Paxillin facilitates timely neurite initiation on soft-substrate environments by interacting with the endocytic machinery

Ting-Ya Chang[1], Chen Chen[1], Min Lee[1], Ya-Chu Chang[1], Chi-Huan Lu[1], Shao-Tzu Lu[1], De-Yao Wang[1], Aijun Wang[2], Chin-Lin Guo[3], Pei-Lin Cheng[1]*

[1]Institute of Molecular Biology, Academia Sinica, Taipei, Taiwan; [2]Surgical Bioengineering Laboratory, Department of Surgery, University of California, Davis, Davis, United States; [3]Institute of Physics, Academia Sinica, Taipei, Taiwan

**Abstract** Neurite initiation is the first step in neuronal development and occurs spontaneously in soft tissue environments. Although the mechanisms regulating the morphology of migratory cells on rigid substrates in cell culture are widely known, how soft environments modulate neurite initiation remains elusive. Using hydrogel cultures, pharmacologic inhibition, and genetic approaches, we reveal that paxillin-linked endocytosis and adhesion are components of a bistable switch controlling neurite initiation in a substrate modulus-dependent manner. On soft substrates, most paxillin binds to endocytic factors and facilitates vesicle invagination, elevating neuritogenic Rac1 activity and expression of genes encoding the endocytic machinery. By contrast, on rigid substrates, cells develop extensive adhesions, increase RhoA activity and sequester paxillin from the endocytic machinery, thereby delaying neurite initiation. Our results highlight paxillin as a core molecule in substrate modulus-controlled morphogenesis and define a mechanism whereby neuronal cells respond to environments exhibiting varying mechanical properties.
DOI: https://doi.org/10.7554/eLife.31101.001

*For correspondence:
plcheng@imb.sinica.edu.tw

Competing interests: The authors declare that no competing interests exist.

## Introduction

Environmental rigidity affects cell morphogenesis and development. For example, mesenchymal cells develop long protrusions and spread when they are cultured on glass coverslips or plastic plates (*Lo et al., 2000*), but fail to do so on soft substrates ($E < 5$ kPa) in cell culture. Likewise, rigid substrates ($E > 20$ kPa) facilitate differentiation of mesenchymal stem cells into osteogenic cells (*Engler et al., 2006*). In fact, the behavior of mesenchymal cells reflects the response of most migratory/adherent cells. By contrast, newborn neurons, which naturally reside in extremely soft tissue environments ($E = 0.1–1$ kPa) such as brain (*Christ et al., 2010*; *Georges et al., 2006*), behave differently. Soft mechanical properties generally favor neuronal differentiation (*Engler et al., 2006*; *Saha et al., 2008*) and maturation (*Balgude et al., 2001*; *Fawcett et al., 1995*; *Leach et al., 2007*). In particular, neurons extend long protrusions known as neurites toward softer surfaces when cultured on substrates showing a stiffness gradient (*Sundararaghavan et al., 2009*). In vivo evidence for the inverse durotaxis of neuronal cells has been provided recently using retinal ganglion cells of *Xenopus* embryos, which exhibited a noticeable pattern of stiffness gradients within the optic tracts toward their targets in the optic tectum (*Koser et al., 2016*). Such distinct behavior plays a crucial role in establishing projections and dendritic territory. Although transduction mechanisms governing the development and behavior of migratory cells on rigid substrates in cell culture are well characterized (*Aragona et al., 2013*; *Dupont et al., 2011*), mechanisms underlying how neurons behave differently from migratory cells on soft substrates remain elusive.

In general, development of cell shape depends on coordination of dynamic membrane activities such as endocytosis, exocytosis, or adhesion with cytoskeletal mechanics. For adherent cells grown on matrices, development of cell shape primarily relies on integrin-mediated adhesions, which recruit molecules such as paxillin and vinculin (*Dumbauld et al., 2013*; *Humphries et al., 2007*; *Schaller, 2001*; *Turner, 2000*) to adhesive anchorages and to generate tensile forces (*Carisey et al., 2013*; *Desmoulière et al., 2005*). The morphogenetic differences between neurons and migratory cells in soft environments suggest that mechanisms other than integrin-mediated adhesion may function in neurite initiation. Potential candidates are changes in the gene expression and cell signaling that govern endocytosis, which functions not only in membrane dynamics but also in integrin internalization and focal adhesion disassembly (*Caswell et al., 2008*; *Du et al., 2011*; *Itofusa and Kamiguchi, 2011*; *Nishimura and Kaibuchi, 2007*; *White et al., 2007*). Furthermore, endocytosis is required to activate and recruit the neuritogenic signal molecule Rac1 (*Palamidessi et al., 2008*) to the membrane, an activity that enhances formation of cell protrusions through actin filament polymerization (*Hall, 1998*; *McMahon and Boucrot, 2011*; *Merrifield and Kaksonen, 2014*).

Neurite initiation is the very first step of a single neuron toward neuronal networking. To fully appreciate the role of soft environments in neuronal development, ranging from cell fate to cell shape, it is challenging but vital to determine the underlying mechanism responsible for the spatiotemporal control of neurite initiation in the embryonic brain. To investigate mechanisms driving neurite initiation in soft tissue environments, we cultured embryonic rat primary hippocampal neurons on hydrogels of various elastic moduli and monitored spatiotemporal patterns of neurite initiation and corresponding changes in gene expression. We observed a bistable pattern of neurite initiation associated with altered expression of genes encoding components of the endocytic machinery. In the absence of neurite-promoting factors, endocytosis was required for cells to form the morphological precursors of neurites, that is, segmented lamellipodia. We identified paxillin as a key protein that directly associates with either the adhesion protein vinculin or the F-BAR-containing endocytic factor CIP4. When grown on soft substrates, cells expressed high levels of paxillin associated with the endocytic machinery, which in turn upregulated Rac1 activity to promote neurite formation and elevate expression of proteins of the endocytic machinery as part of a positive feedback loop. By contrast, cells grown on rigid substrates developed numerous adhesions, which sequestered paxillin from the endocytic machinery and delayed neurite initiation. Using genetic profiling and biochemical approaches, we show that paxillin-mediated endocytosis and formation of adhesions constitute a bistable switch to control neurite initiation in a substrate modulus-dependent manner.

## Results

### Bistable substrate modulus-dependent behavior in neurite initiation

We used polymerized hydrogels to define the mechanisms underlying neurite initiation in soft environments. Gels of three elastic moduli—0.1, 1 and 20 kPa—were engineered and verified by atomic force microscopy (see Materials and methods, and *Figure 1—figure supplement 1A–D*). Culturing hippocampal cells isolated from embryonic day 17.5 (E17.5) rat brain on these compliant gels can selectively enrich the population of neurons by up to 80% and minimize growth of glial cells (*Figure 1—figure supplement 1E*). After seeding cells onto gels and incubating them for either 5 or 16 hr in regular neurobasal medium, we assessed potential morphogenetic changes in cells over time. Upon initial gel contact, cells formed a uniform lamellipodial extension (*Figure 1A1*, and *Figure 1—figure supplement 2A,B*), which became segmented into multiple, separated lamellipodia if the gel was sufficiently soft (see *Figure 1—figure supplement 2A–D*). Lamellipodia are well-known morphological precursors of neurites (*Figure 1—figure supplement 2*; *Dehmelt et al., 2003*). However, we observed that neurites formed only if their preceding lamellipodia occupied less than a third of the entire cell periphery; lamellipodia occupying >1/3 of the periphery failed to form neurites even at the 16 hr time-point (*Figure 1A2–A4*, and *Figure 1—figure supplement 2A–D*). Therefore, we defined each lamellipodium with <1/3 occupancy of the entire cell periphery as a 'segmented' lamellipodium (SL), and those occupying >1/3 as a 'broad' lamellipodium (BL). In most, if not all, cases of the 5 hr cultures, a single cell could not have both SL and BL.

We identified a bistable substrate modulus-dependent behavior by measuring the distribution of SL and BL over various substrates of varying stiffness (*Figure 1A3*, and *Figure 1—figure*

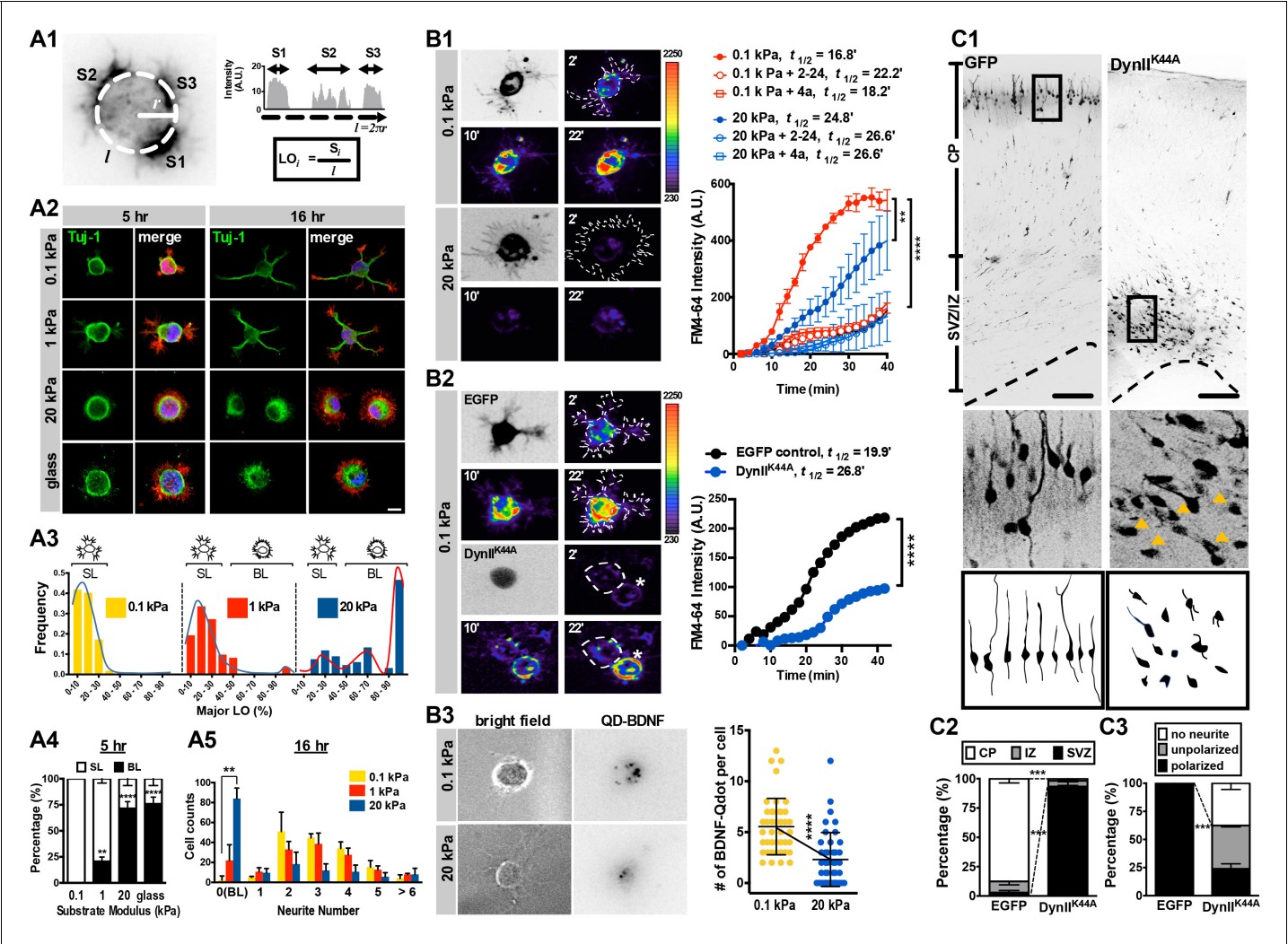

**Figure 1.** Morphology and endocytic activity of neurons grown on substrates of varying stiffness. (**A**) Substrate modulus-dependent biphasic distribution of lamellipodium occupancy. (**A1**) Representative intensity profile of phalloidin-stained hippocampal neurons on a soft hydrogel ($E$ = 0.1 kPa). Lamellipodium occupancy (LO) of each phalloidin-positive segment is calculated by the formula shown in the box at bottom right. (**A2**) Representative images of neurons grown on substrates for 5 hr or 16 hr and stained with phalloidin (Red), DAPI (Blue), and antibodies against Tuj-1 (Green), as indicated. Scale bar: 20 µm. (**A3**) Histograms of LO distributions at 5 hr showing two distinct patterns in 0.1, 1, or 20 kPa cultures (n > 68 cells for each experiment). Simplified drawings above histograms illustrate the typical segmented lamellipodium ('SL') and broad lamellipodium ('BL') phenotypes in 5 hr neuronal cultures, as indicated. (**A4 and A5**) Histograms summarizing the percentages of lamellipodium phenotypes seen in 5 hr cultures (**A4**) and the distributions of neurite numbers (**A5**) in 16 hr cultures. Data represent mean ±SEM (n > 3 independent experiments; 150 cells for each culture; *p<0.05; **p<0.01; ****p<0.0001; one way ANOVA with *Dunnett's post hoc* test). (**B**) Enhanced membrane endocytosis in 0.1 kPa cultures. Time-lapse images (20 frames; 2 min intervals) of neurons isolated from E17.5 rat cortices transfected in utero at E16 without (**B1**) or with (**B2**) IRES constructs harboring control EGFP and/or a dominant-negative dynamin II mutant (DynII^K44A), cultured on 0.1 kPa or 20 kPa gels for 5 hr, followed by endocytosis assay in the presence or absence of the endocytosis inhibitors Dynole 2–24 or Dyngo 4a, as indicated. Dashed lines surround the region of interest (ROI) in quantitative FM4-64 measurements. Asterisks in B2 mark non-transduced neighboring cells. Graph at right summarizes the accumulation curves of FM4-64 signal (±SEM, n > 3 independent experiments, 10–20 cells per group, normalized to t = 0 value; ****p<0.0001; two-way RM ANOVA with *Dunnett's post hoc* test), which reflects the rate of FM4-64 uptake at different time-points after FM4-64 loading. The data were fitted to a single exponential to determine the time ($t_{1/2}$) required to reach half of the plateau value. (**B3**) Representative images of neurons plated on 0.1 kPa or 20 kPa gels overnight, followed by incubation of quantum dot-conjugated brain-derived neurotrophic factor (QD-BDNF) for 5 hr. Dot plot showing that 0.1 kPa cultures exhibited a significantly higher level (n = 39–46 cells from three independent experiments; **p<0.001; ****p<0.0001; one-way ANOVA with *Dunnett's post hoc* test) of QD-BDNF internalization (as reflected by a ~2-fold greater quantity of intracellular QD-BDNF) than 20 kPa cultures. (**C**) Expression of DynII^K44A prevents neurite formation and cortical neuron migration in vivo. (**C1**) Fluorescence images of P0 rat cortices transfected in utero at E16 with IRES constructs harboring control EGFP and/or DynII^K44A. The middle panels show 16x magnifications of boxed regions of the corresponding P0 cortex in the top panels. The bottom panels show sample tracings of 2D projections from confocal images of typical cortical neurons in the corresponding P0 cortex. Bar, 100 µm. (**C2 and C3**) Histograms showing the localization (**C2**) and the percentage (**C3**) of transfected

*Figure 1 continued on next page*

*Figure 1 continued*

cortical neurons exhibiting unipolar/bipolar polarized processes ('polarized'), multiple short neurites without a long tailing process ('unpolarized'), or no process ('no neurite'; arrowheads in the middle panels) in the cortical plate ('CP') or subventricular zone/intermediate zone ('SVZ/IZ') regions. Datasets (mean ±SEM, n > 150 cells per cortex, >5 cortices each; ***p<0.001, multiple *t* test) showing significant differences are marked.

DOI: https://doi.org/10.7554/eLife.31101.002

The following figure supplements are available for figure 1:

**Figure supplement 1.** Hippocampal neurons cultured on polyacrylamide hydrogels of varying stiffness.

DOI: https://doi.org/10.7554/eLife.31101.003

**Figure supplement 2.** Differential lamellipodium phenotypes of neurons grown on hydrogels.

DOI: https://doi.org/10.7554/eLife.31101.004

**Figure supplement 3.** Differential gene expression pattern of neurons grown on hydrogels.

DOI: https://doi.org/10.7554/eLife.31101.005

*supplement 2E,F*). Cells could be segregated into two populations, that is, those with or without SL formation, in a substrate modulus-dependent manner. For example, in 5-hr soft-gel cultures (0.1 kPa), nearly 100% of cells formed SL (*Figure 1A4, A5*, and *Figure 1—figure supplement 2E*), whereas most cells (~71.5%) on stiff-gel cultures (*E* = 20 kPa) or coverslips (glass) failed to form SL and instead formed a single BL (*Figure 1A4*, and *Figure 1—figure supplement 2E,F*). In 16 hr cultures, the SL of most (~98%) cells on soft gels had become neurites, whereas more than half (~57.3%) of the cells on stiff gels possessed BL and lacked neurites (*Figure 1A5*). Such bistable behavior suggests the existence of two distinct states in early stages of neuritogenesis; a 'segmentable state' in which cells can form segmented lamellipodia as morphological predecessors of neurites, and a 'non-segmentable state' in which cells retain a spreading morphology with broad lamellipodia. The fact that cells form neurites on soft gels in the absence of neurite-promoting growth factors suggests that culturing cells on environments with elastic moduli <1 kPa, which match the mechanical properties of brain (*E* = 0.1—1 kPa; (*Christ et al., 2010*; *Georges et al., 2006*), is sufficient to initiate neurite formation. Moreover, surface laminin was detectable on 0.1 kPa, 20 kPa and glass surfaces, and it was comparable following 5 and 16 hr incubation periods in neuronal culture media (see *Figure 1—figure supplement 1D*), ruling out a difference due to coating. Note that when culturing cells on coverslips, lower laminin/PLL concentrations may be required for proper laminin/integrin interaction to overcome the influences of high substrate rigidity (please see *Appendix 1—figure 1*).

## Substrate modulus-dependent morphogenesis is associated with distinct cell signaling and gene expression patterns

To ascertain whether changes in gene expression are associated with segmentable or non-segmentable states, we undertook genetic profiling of cells grown on soft or stiff substrates at 5 or 16 hr time-points. Using mRNA microarray and Gene Ontology analyses, we identified 114 differentially expressed mRNA transcripts in cells grown on 0.1 kPa and 20 kPa gels at the 5 hr time-point (filtering criteria: fold change ≥1.5, p<0.05; *Figure 1—figure supplement 3A*, and *Appendix 1—table 1*, *2*). Among them, 66 were upregulated in 0.1 kPa gel cultures and the remainder were downregulated (compared to cells grown on stiffer substrates). Notably, three genes—*Cltc*, *Dab2*, and *Myo6*, all of which function in clathrin-mediated endocytosis—were co-upregulated in 0.1 kPa gel cultures at the 5 hr time-point (*Figure 1—figure supplement 3B*), suggesting that cells enter a high endocytic state on a soft substrate (*E* = 0.1 kPa). By contrast, several adhesion factors, including *Vcl*, *Nrcam*, *Robo2* and *Cdh11*, were relatively upregulated in 20 kPa gel cultures at 5 hr and/or 16 hr compared to cells grown on softer substrates (*Figure 1—figure supplement 3B*).

To verify that changes in gene expression associated with endocytosis or adhesion were correlated with phenotypic changes, we performed single molecular RNA fluorescent in situ hybridization (smFISH, see Materials and methods for details), which enables mRNA quantitation in two groups of single cells based on their morphological signatures, that is, BL or SL (*Figure 1—figure supplement 3C*). For this analysis, we examined nine genes involved in three different types of cellular activities: endocytosis, adhesion, and the Hippo-YAP pathway. All these pathways regulate membrane mechanics under certain conditions. In 5 hr cultures, we found that SL cells expressed relatively lower (2.3- to 10-fold) and higher (1.2- to 4-fold) levels of adhesion-related (*Vcl, Cdh11 and Robo2*) and

endocytic (*Cltc*, *Dab2*, and *Myo6*) mRNA, respectively, compared to those of BL cells (*Figure 1—figure supplement 3C*). However, no significant differences were observed in levels of the Hippo-YAP-related (*DKK1* and *CYR61*) mRNA between SL and BL cells. Although fold-changes in gene expression vary across signal detection methods (*Figure 1—figure supplement 3*), these findings are consistent with the microarray data and suggest that cells with segmented lamellipodia undergo preferential gene expression of endocytic- but not adhesion-related genes.

## Elevated endocytic activity is associated with soft substrate environments and is critical for neurite development in vivo

To ascertain whether neurons enter an endocytic-dominant state upon contact with soft environments, we measured endocytic activity in cells grown on soft or stiff gels. To assess endocytosis, first we monitored uptake of the lipophilic dye FM4-64 over a 40 min period (at 2 min intervals) in 5 hr cultures (*Figure 1B1*). The rate of FM4-64 uptake was reflected by the time required to reach half of the maximum FM4-64 intensity measured in cells, defined as $t_{1/2}$. We observed that cells grown on 0.1 kPa gels exhibited a ~1.6-fold greater FM4-64 intensity at the end-point of the measurements (t = 40 min), and a ~1.7-fold greater uptake rate [$t_{1/2\ (0.1\ kPa)}$ v.s. $t_{1/2\ (20\ kPa)}$=16.8 min v.s. 24.8 min] than cells grown on 20 kPa gels (*Figure 1B1*). Robust dye uptake was pharmacologically and genetically abolished by applying inhibitors of dynamin GTPase and by ectopically expressing a dominant-negative Dynamin II mutant (DynII[K44A]; (*Damke et al., 1994*; *Herskovits et al., 1993*) in 0.1 kPa cultures, respectively (*Figure 1B1, B2*). These results suggest that dye uptake is primarily endocytosis-associated. Note that constructs encoding DynII[K44A] were in utero eletroporated (IUE, (*Saito and Nakatsuji, 2001*) into the developing brain of E16 rat embryos. Cultures were prepared 24 hr after electroporation to allow for sufficient expression of DynII[K44A]. To confirm that levels of legend-mediated endocytosis were elevated in neurons grown on soft substrates, we then quantified invagination of quantum dot-conjugated brain-derived neurotrophic factor (denoted as QD-BDNF and exhibiting a 1:1 BDNF:QD conjugation ratio; see Materials and methods) in 16 hr cultures (*Figure 1B3*). We observed an average invagination rate of ~7 dots per cell in 0.1 kPa cultures over the first 5 hr after bath application of QD-BDNF. That rate was significantly decreased to two dots per cell in 20 kPa cultures (*Figure 1B3*), and comparable results were seen following dynamin inhibition (*Figure 1B2*). These findings suggest that cells grown on soft substrates undergo a high level of endocytosis.

These findings prompted us to examine the functional relevance of endocytic activity on neurite formation in vivo. To do so, we used in utero electroporation to express constructs encoding a dominant-negative Dynamin II mutant (DynII[K44A]) in a subpopulation of neural progenitor cells (*Figure 1C*). To aid observations of morphogenetic phenotypes, brain slices were obtained from newborn (P0) rat cortex when most newly differentiated neurons had extended their neurites and migrated towards the cortical plate. Nearly all (~100%) control cortical neurons (i.e. those solely expressing marker protein) exhibited a neurite-bearing, polarized morphology (*Figure 1C1, C3*), with dendritic arbors being oriented toward the pial surface and the axon being oriented radially in the cortical plate. By contrast, cortical neurons expressing DynII[K44A] exhibited an apparent migration defect (*Figure 1C1, C2*), with a high percentage of cells having a non-polarized morphology without extension of minor processes (termed 'no neurite') and a reduced percentage of cells exhibiting a polarized morphology (unipolar or bipolar) (*Figure 1C3*). Thus, a normal level of endocytic reactions is required for proper neurite formation and radial migration of newly generated cortical neurons; two tightly linked events during neuronal development in vivo.

## The endocytic-dominant state on soft substrates favors Rac1 signaling and neurite formation

Endocytosis enhances membrane translocation and Rac1 activation (*Palamidessi et al., 2008*) and, although in opposing manners, both RhoA and Rac1 function in neurite formation (*Aoki et al., 2004*; *Da Silva et al., 2003*; *Govek et al., 2005*; *Machacek et al., 2009*); reviewed in *Guilluy et al. (2011)*. Given the dominance of endocytosis events on soft gels, we wondered whether the substrate modulus governed Rac1 activity. Using an active Rac1 pulldown assay, immunofluorescent staining of Rac1 or active Rac1, and the FRET biosensor Rac1-2G (*Fritz et al., 2013*; *Fritz et al., 2015*), we observed that, unlike RhoA, Rac1 activity decreased with increasing stiffness of the substrate in 5 hr neuronal cultures (*Figure 2—figure supplement 1A*). In addition, active Rac1

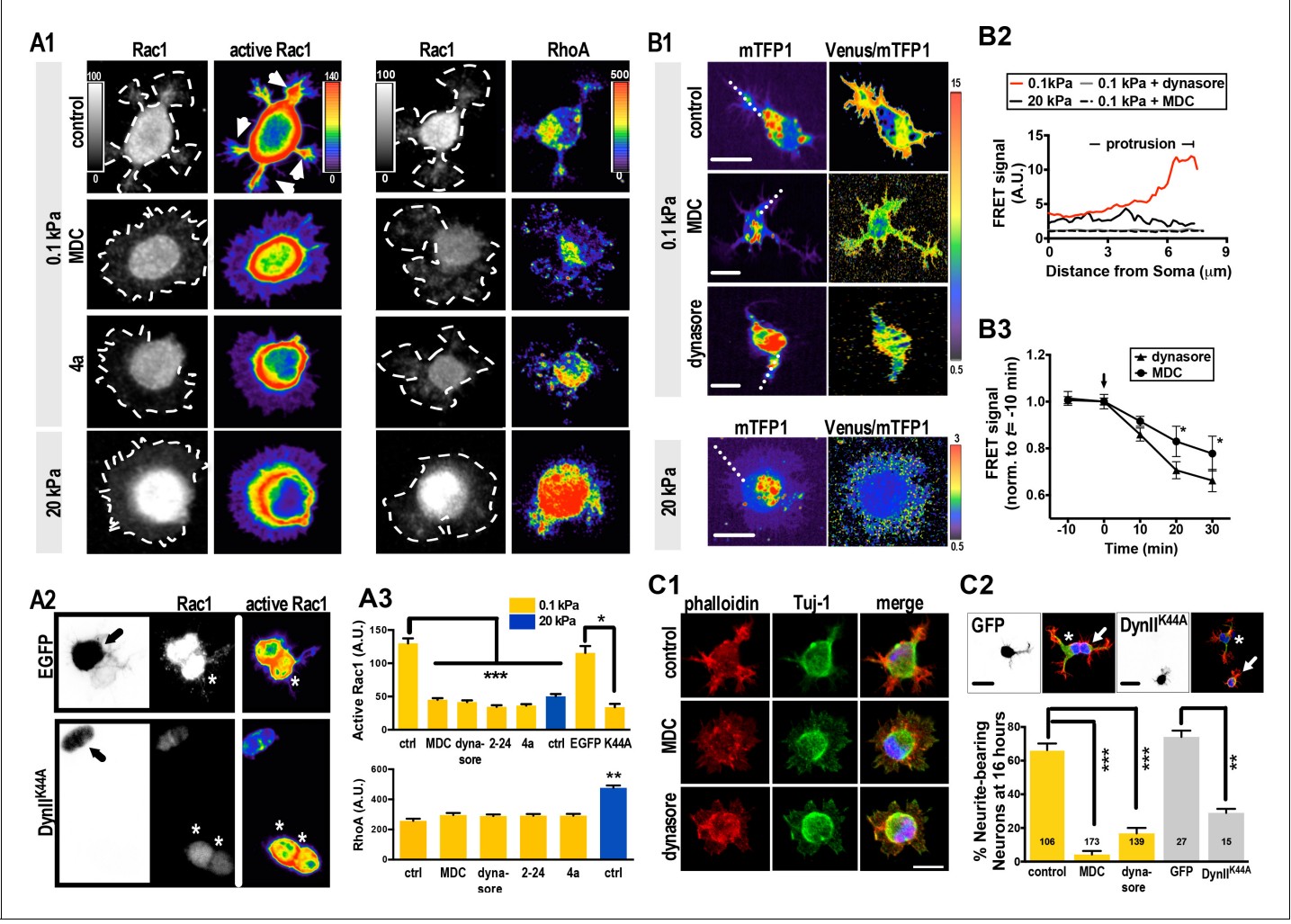

**Figure 2.** Substrate modulus-dependent Rac1 increase and lamellipodium segmentation require endocytic activity. (**A**) Inhibition of endocytosis suppresses Rac1 activity in 0.1 kPa cultures. (**A1–A3**) Representative images of 5 hr neuronal cultures treated with the endocytosis inhibitors dynasore (50 μM, 1 hr), Dynole 2–24 (2–24, 2 μM, 1 hr), Dyngo 4a (4a, 2 μM, 1 hr) or monodansylcadaverine (MDC, 100 μM, 1 hr) and immunostained with antibodies against Rac1, GTP-bound Rac1 ('active Rac1'; A1, left panel), or RhoA (A1, right panel). The same staining procedure was applied to the IUE neuron (A2, arrow) expressing control EGFP and/or DynII[K44A] as shown in A2. Asterisks mark non-transduced neighboring cells. Intensities of active Rac1 and RhoA correspond to linear pseudocolor maps. (**A3**) Histograms, all from experiments similar to those described above, showing that endocytosis inhibition significantly decreases levels of total Rac1 and active Rac1 (but not RhoA) at lamellipodial protrusions (at 5 hr) in 0.1 kPa cultures. Data represent mean intensity ±SEM (n > 50 per group from three independent experiments; ***p<0.001; one-way ANOVA with *Dunnett's post hoc* test). (**B**) Representative FRET-based imaging of hippocampal neurons transfected with a FRET indicator for GTP-bound Rac1 (Rac1-2G; see Materials and methods). (**B1**) Images of monomeric Teal fluorescent protein 1 (mTFP1) and FRET signals (presented as pseudocolor maps in a linear scale) acquired from neurons plated on 0.1 kPa gels in the presence or absence of dynasore (50 μM) or MDC (100 μM) for 1 hr. Bar, 20 μm. (**B2**) Traces depict active Rac1 levels at protrusions, measured across the dashed line as shown in B1, and indicated by FRET signal calculated as the ratio of Venus to mTFP1 fluorescence [F(Venus)/F(mTFP1)]. (**B3**) Quantitative measurement of FRET signals (±SEM, n = 5 cells for each group; *p<0.05, compared to time = 0 value; multiple *t* tests) before and after bath application of 50 μM dynasore or 100 μM MDC. Arrow marks drug addition (time = 0). (**C**) Inhibition of endocytosis decreases the probability of lamellipodium segmentation and delays neurite initiation on soft gels. (**C1**) Representative fluorescent images of neurons plated on 0.1 kPa gels for 5 hr in the absence or presence of indicated endocytosis inhibitors. Scale bar: 20 μm. (**C2**) Representative fluorescent images of transfected IUE neurons (arrow) expressing control EGFP and/or a dominant-negative dynamin II mutant (DynII[K44A]), cultured on 0.1 kPa gels for 16 hr, and stained with phalloidin (Red), DAPI (Blue), and antibodies against Tuj-1 (Green). Asterisks mark non-transduced neighboring cells. Scale bar: 20 μm. Histograms summarize the percentages (±SEM; ***p<0.001, one way ANOVA with *Dunnett's post hoc* test) of 16 hr neurons bearing neurites in the presence or absence of dynasore (50 μM, 5 hr treatment) or MDC (100 μM, 5 hr treatment), as indicated.
DOI: https://doi.org/10.7554/eLife.31101.006

The following figure supplement is available for figure 2:

**Figure supplement 1.** Increased Rac1 activity in neurons grown on 0.1 kPa gels.

*Figure 2 continued on next page*

*Figure 2 continued*

DOI: https://doi.org/10.7554/eLife.31101.007

accumulated in SL and the tips of growing neurites, whereas there was no accumulation in BL (*Figure 2A1, A3*). Consistently, ratiometric FRET imaging confirmed accumulation of active Rac1 at segmented neuronal lamellipodia on soft gels (0.1 kPa), with FRET signal decreasing from the leading edge to the cell body (*Figure 2B1, B2*). Moreover, either expression of DynII$^{K44A}$ or treatment of the 0.1 kPa cultures with inhibitors of endocytosis significantly blocked accumulation of active Rac1 (*Figure 2A1–A3*). These results suggest that Rac1 accumulation requires endocytosis.

Previous studies have shown that Rac1 activation induces cell protrusions through actin filament polymerization (reviewed in [*Hall, 1998*; *Ridley, 2011*]). To ascertain whether Rac1 increase and accumulation on soft gels (0.1 kPa) was responsible for neurite initiation, we treated cells grown on soft gels with the Rac1 inhibitor NSC 23766 immediately after plating and assessed phenotypes at the 5 hr time-point, which resulted in the onset of lamellipodial segmentation being significantly delayed (see 48 hr time-point in *Figure 2—figure supplement 1B*). In support of the idea that Rac1 accumulation requires endocytosis (*Figure 2A*), inhibition of endocytosis in soft-gel cultures for 1 hr also significantly increased manifestation of the BL phenotype at the 5 hr time-point (*Figure 2C1*), with >90% of cells exhibiting sparsely distributed and less activated Rac1 and broad lamellipodia that resembled those seen on 20 kPa gels (*Figure 2A,B*). The number of neurite-bearing neurons was also decreased to ~13% compared to the untreated control (*Figure 2C2*). In contrast, bath-application of Y27632 (an inhibitor of Rho-associated protein kinase, ROCK) or the PI3k inhibitor LY294002 to 0.1 kPa cultures did not alter the onset of lamellipodial segmentation (*Figure 2—figure supplement 1B*). These results suggest that culturing cells on a soft substrate is sufficient to promote a neuritogenic state through endocytosis-associated Rac1 activation.

## Model for competitive recruitment of paxillin to divergent cellular machineries as part of a bistable switch

In many biological systems, selection between two distinguishable states requires changes in gene expression, which are then amplified via a bistable switch (reviewed in [*Ferrell, 2002*; *Smits et al., 2006*]). Our results suggest that newborn neurons possess two morphological states: one with the ability to form segments and neurites and the other without. These states are associated with specific genetic changes that are governed by substrate stiffness, cell contractility, and endocytic activity (*Figure 1* and *Figure 2*). To account for both endocytosis and adhesion, here we assume a simple 'sequestering' mechanism whereby a factor/switcher required for both endocytosis and cell-substrate adhesion exists (mathematical model and figure shown in Appendix). To identify the molecular switch, we evaluated by western blot analysis the protein levels of candidate molecules functioning in adhesion or the endocytosis machinery. In 5 hr cultures on stiff substrates (20 kPa gels or glass), cells expressed higher levels of adhesion factors (such as integrin β1; for the level of active integrin, please see *Appendix 1—figure 3*) than endocytic factors (*Figure 3A,B*). Although pFAK levels increased slightly on stiff substrates, the changes were not statistically significant. By contrast, cells grown on soft gels (0.1 kPa and 1 kPa) expressed higher levels of factors associated with clathrin-mediated endocytosis, including myosin VI, clathrin heavy chain (CHC) and Cdc42 interacting protein 4 (CIP4) (*Figure 3A,B*). Among the candidates of interest, we also evaluated the scaffolding protein paxillin, which is known to be an integral component of focal adhesions that bind vinculin and FAK (*Schaller, 2001*; *Turner et al., 1990*). We observed that expression levels of paxillin and phosphorylated paxillin in cultured neurons did not match the profiles of other adhesion factors, but instead matched the profiles of endocytic factors (*Figure 3A and B*). Further analysis of relative paxillin expression in 5 hr cultures indicated that cells grown on soft substrates (0.1 kPa gels) expressed significantly higher levels of paxillin than those grown on stiff substrates (*Figure 3B*). These observations strongly suggest that paxillin is associated with the endocytic machinery in soft cultures and may serve as the proposed switcher.

Evidence for a role of paxillin in endocytosis is limited (but see [*Duran et al., 2009*]). To examine this possibility, we assessed if paxillin is associated with endocytic vesicles using a detergent-free membrane flotation assay of embryonic rat cortex (with cardiac tissue used for comparison). We

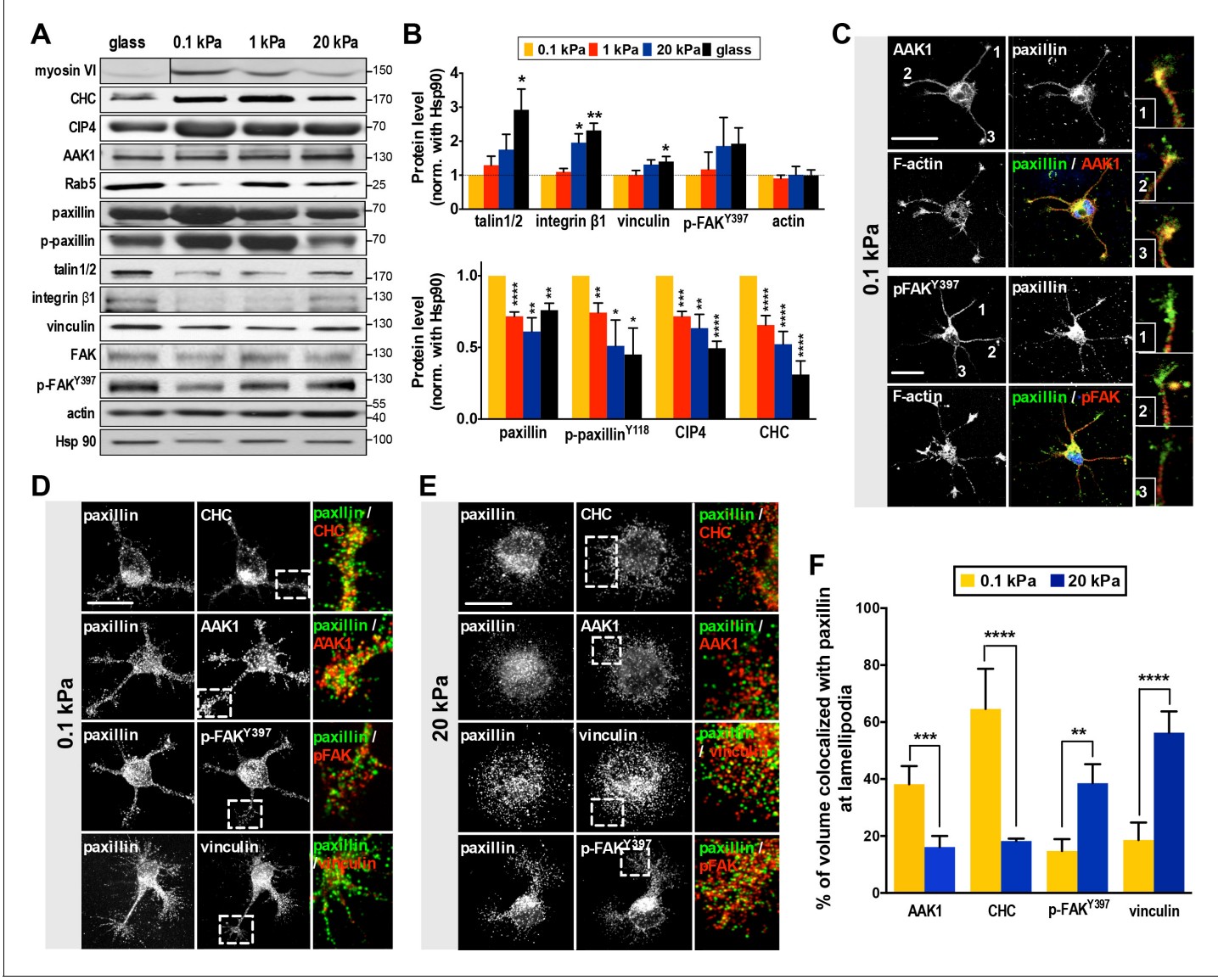

**Figure 3.** Substrate modulus-dependent expression of paxillin and its association with endocytic and adhesion machineries. (A) Representative western blots of endocytic or adhesion molecules. Cell lysates obtained from cortical neurons grown on substrates with differing stiffness were subjected to immunoblotting with antibodies to indicated proteins. (B) Summary histograms, all from experiments similar to that described in A, showing greater (>2 fold) abundance of paxillin, phospho-paxillin$^{Y118}$ (p-paxillin), myosin VI (myo6), CIP4 and clathrin heavy chain (CHC) proteins in neurons grown on soft substrates (0.1 and 1 kPa) compared to those grown on stiff substrates (20 kPa and glass). Note the inverse expression of endocytic factors and adhesion molecules (i.e., talin1/2, integrinβ1, vinculin and p-FAK$^{Y397}$). Data represent mean ±SEM (n ≥ 3, normalized to control actin; compared to 0.1 kPa cultures; *p<0.05; **p<0.01; ***p<0.001; t-test). (C–F) Paxillin co-localizes with the endocytosis complex at neuronal growth cones on soft substrates. (C) Representative confocal images of 16 hr neurons on 0.1 kPa gels co-immunostained with antibodies against paxillin (Green in merge panel), adaptor-associated kinase1 (AAK1, Red in merge panel), p-FAK$^{Y397}$ (Red in merge panel), or F-actin, as indicated. Right panels show the region of interest ROI (marked by numbers) of neurite tips represented at higher magnification. Bar: 20 μm. (D and E) Similar to C, except the resolution of images has been enhanced (~1.7X higher) using Airyscan. Note that surface rendering was applied at neurite tips to more clearly show co-localization (yellow) of paxillin with indicated factors. Bar: 20 μm. (F) Histograms, all from experiments similar to those described in d and e, summarizing percentages of paxillin co-localized with indicated endocytic or adhesion factors on hydrogels. Data represent percentages (±SEM, n = 15 neurons for each set of experiments; ROI, 5 × 5 μm within one lamellipodium; **p<0.01; ***p<0.001; ****p<0.0001; t test).

DOI: https://doi.org/10.7554/eLife.31101.008

The following video and figure supplements are available for figure 3:

**Figure supplement 1.** Paxillin associates with free-floating vesicles in rat embryonic brain lysate.
DOI: https://doi.org/10.7554/eLife.31101.009

**Figure supplement 2.** Ectopically expressed paxillin displays predominantly retrograde motility.

*Figure 3 continued*

DOI: https://doi.org/10.7554/eLife.31101.010

**Figure 3–video 1.** Fast transport of paxillin-mCherry.

DOI: https://doi.org/10.7554/eLife.31101.011

found >30% of paxillin from brain lysates co-fractionated with the endocytosis factors Rab5 and CHC (in floating membrane fractions), whereas co-fractionation was <5% in cardiac lysates (*Figure 3—figure supplement 1*), suggesting that a significant proportion of paxillin is associated with endocytic vesicles in neurons. Live cell imaging also revealed a long-range, predominantly retrograde transport of paxillin-mCherry from axonal tips (*Figure 3—figure supplement 2*; *Figure 3—video 1*); a behavior similar to that seen in neuronal signaling endosomes (*Cosker and Segal, 2014*; *Yap and Winckler, 2012*). Next, we wondered whether paxillin preferentially binds to endocytic rather than adhesion factors in the segmentable state. Using 5 hr soft-gel cultures, we conducted immunostaining of endocytosis-associated kinase AAK1 (*Conner and Schmid, 2002*; *Ricotta et al., 2002*) or adhesion kinase p-FAK$^{Y397}$. A significant proportion (~39%) of paxillin staining colocalized with that of AAK1 rather than p-FAK$^{Y397}$ on soft gels (0.1 kPa; *Figure 3C*). This outcome was quantitatively confirmed by confocal microscopy with enhanced resolution (~1.7X greater using Airyscan). In cells exhibiting an SL phenotype on soft gels (0.1 kPa), enhanced resolution confocal microscopy also revealed that a significant portion (>40%) of paxillin appeared in a punctate pattern and colocalized with the endocytic factors CHC or AAK1 (Pearson's coefficient >0.75) rather than adhesion factors p-FAK$^{Y397}$ or vinculin (Pearson's coefficient <0.45) (*Figure 3D,F*). Interestingly, paxillin preferentially associated with p-FAK$^{Y397}$ and vinculin in cells exhibiting BL on stiff gels (20 kPa; *Figure 3E, F*). Together, these observations suggest that paxillin functions in both endocytic and adhesion activities.

## Endocytic factor CIP4 and adhesion factor vinculin are potential competitors for paxillin binding

Validation of our model requires that a molecular switch binds to endocytic and adhesion machineries in a competitive and substrate modulus-dependent manner. To determine whether paxillin meets these criteria, we tested its binding to the endocytic machinery using liquid chromatography-tandem mass spectrometry (LC−MS/MS) of lysates from rat embryonic brain and in vivo co-immunoprecipitation of cortical neurons cultured on hydrogels. Our analyses identified several endocytic factors that bind to paxillin, including clathrin, dynamin, and CIP4 (*Figure 4A–C*, and *Appendix 1—table 3*). Moreover, the binding affinity of paxillin to these factors increased with substrate compliance, whereas the binding affinity of paxillin to vinculin increased with substrate stiffness (*Figure 4A*). These findings indicate that paxillin can switch its binding preference in a substrate modulus-dependent manner.

Next, we examined whether binding of paxillin to endocytic or adhesion factors is competitive. Structurally, paxillin is an adaptor protein composed of N-terminal LD motifs and C-terminal LIM domains, both of which are important for binding to focal adhesion factors (*Brown et al., 1996*; *Brown et al., 1998*). Among those factors, vinculin exhibits a well-characterized paxillin-binding subdomain or 'PBS' (*Tachibana et al., 1995*; *Wood et al., 1994*). By evaluating potential structural similarities between endocytic factors and the vinculin PBS, we found that the F-BAR domain of CIP4 shares the greatest degree of similarity with the PBS (*Figure 4—figure supplement 1A*). To ascertain whether this domain binds paxillin, we used GST-fused full-length paxillin (GST-PXN$^{FL}$) to pull down bacterially expressed, histidine-tagged full-length (His-CIP4$^{FL}$) or F-BAR domain-deleted (His-CIP4$^{\Delta F\text{-}BAR}$) CIP4. Both proteins were HPLC-purified and biochemically confirmed by mass spectrometry. GST-PXN$^{FL}$ associated with His-CIP4$^{FL}$ but not His-CIP4$^{\Delta F\text{-}BAR}$ (*Figure 4B*), confirming that the F-BAR domain is required for paxillin binding.

Furthermore, using an in vitro GST pulldown assay and domain mapping analysis in HEK293T cell lysates, we found that deletion of the LIM3-4 domain reduced paxillin affinity for vinculin and CIP4 (*Figure 4C,D*). However, the paxillin interaction domains for vinculin and CIP4 do not fully overlap, as vinculin and CIP4 primarily associated with LD motifs (LD1, 2, and/or 4; [*Turner et al., 1999*]) and the LIM domain, respectively (*Figure 4—figure supplement 1*). In addition, deletion of LIM domains

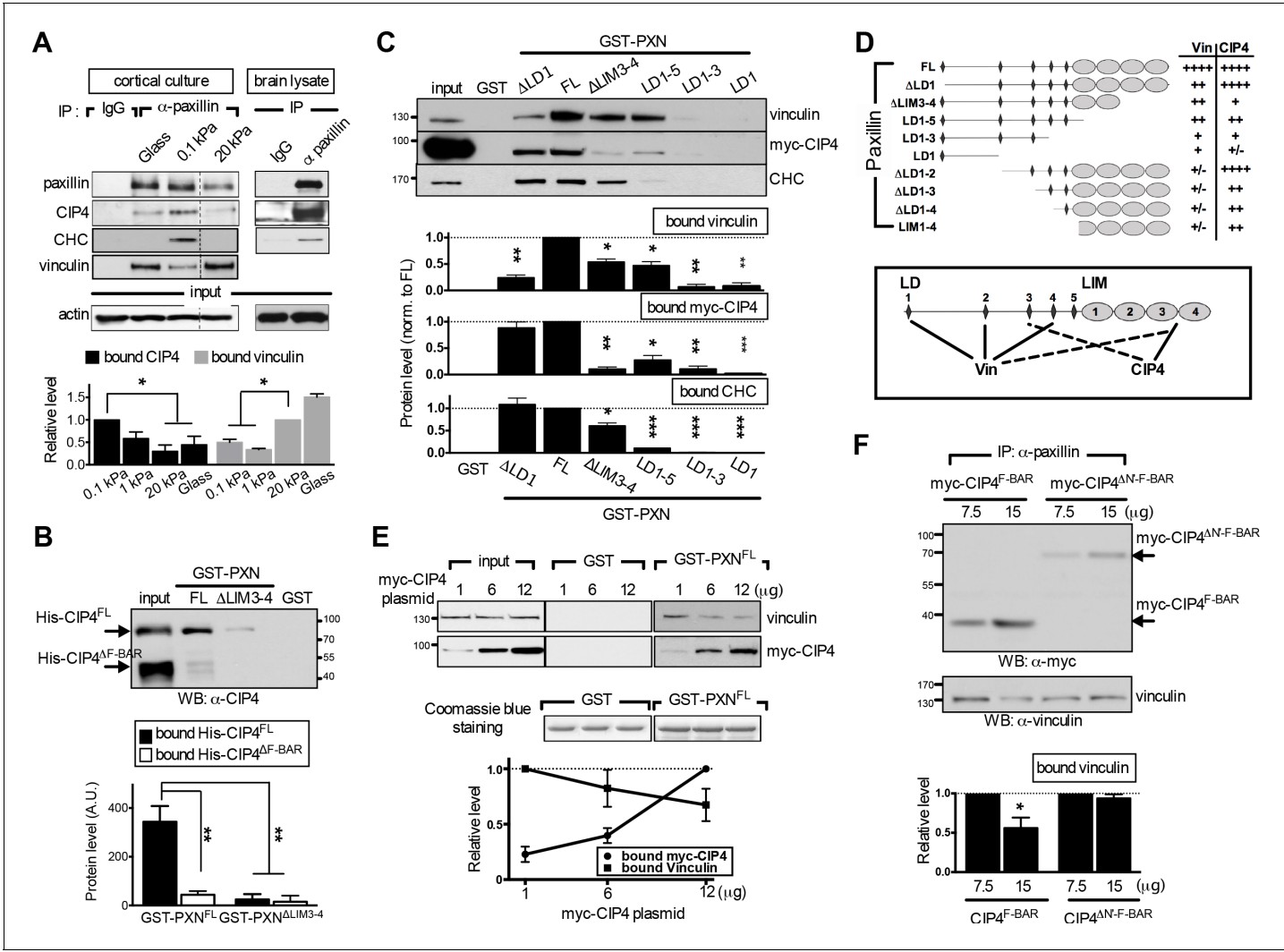

**Figure 4.** The endocytic F-BAR protein CIP4 directly associates with paxillin and competes with vinculin for paxillin binding. (**A**) Paxillin preferentially binds endocytic factors in neurons grown on soft substrates. Paxillin-associated complexes were immunoprecipitated (IP) in lysates made from E17.5 rat brain or from cortical neuronal cultures grown on different substrates using a specific paxillin antibody and were then detected by western blot analysis. Normal rabbit IgG ('IgG') served as a negative control. Histograms show the opposing binding preference of paxillin toward CIP4/CHC or vinculin when grown on soft (0.1 kPa or 1 kPa) versus rigid (20 kPa or glass) substrates. Data represent mean intensity ±SEM (n = 3 independent experiments; *p<0.05; *t*-test). (**B**) Western blot showing direct interaction of paxillin with CIP4. Bacterially expressed His-CIP4 was purified by fast protein liquid chromatography and subjected to a GST pull-down assay using GST-PXN$^{FL}$, GST-PXN$^{\Delta LIM3-4}$ or GST alone. Precipitants were analyzed by western blotting with antibodies specific to CIP4. Histograms summarize protein levels as determined by immunoblotting of full-length (His-CIP4) or F-BAR domain-deleted (His-ΔF-BAR) CIP4 pulled-down by GST-paxillin variants (±SEM, n = 3; normalized to the corresponding GST-PXN$^{FL}$ or GST-PXN$^{\Delta LIM3-4}$ inputs; **p<0.01, *t* test). (**C** and **D**). Mapping of paxillin domains interacting with CIP4 or vinculin. (**C**) GST pull-down and immunoblotting of vinculin, myc-CIP4, and CHC in lysates of myc-CIP4-expressing HEK293T cells. Histograms reflect quantification of levels of proteins pulled-down by GST fusions of full-length ("FL") or LIM domain- and/or LD motif-deleted forms of paxillin, all from experiments similar to those shown in top panels (±SEM, n ≥ 3 independent experiments; *p<0.05; **p<0.01; ***p<0.001; *t* test). (**D**) Schematic of GST fusion proteins used in c. Table summarizing relative CIP4 or vinculin ('Vin') binding by paxillin deletion mutants or full-length protein. Solid lines mark primary sites of interaction, and dashed lines mark accessory interaction motifs for strong binding to vinculin or CIP4. Binding strength relative to full-length paxillin indicated as: '++++' >75% > '+++' >50% > '++' >25% > '+' >5% > '+/-'. (**E**) In vitro protein interaction and competitive binding assays in HEK293T cells transfected with various amounts (1, 6, and 12 μg) of plasmids encoding myc-tagged CIP4 protein (myc-CIP4) and/or control vectors, as indicated. Cell lysates were subjected to a GST pull-down assay with GST-PXN$^{FL}$ or GST alone, and immunoblotted with vinculin and myc antibodies. Line chart depicts averaged protein levels as determined by immunoblotting of CIP4 or vinculin pulled-down by GST-PXN$^{FL}$ (±SEM, n = 4; normalized to band intensity of corresponding GST-paxillin variant). (**F**) In vivo protein interaction and competitive binding assays in HEK293T cells transfected with various amounts (7.5 μg and 15 μg) of plasmids encoding the F-BAR domain ('F-BAR') alone or F-BAR-domain-deleted ('ΔN'-F-BAR') CIP4 and/or control vectors, as indicated. Cell lysates were immunoprecipitated by paxillin antibodies and blotted with myc or vinculin antibodies. Histograms show relative protein levels as determined by immunoblotting of vinculin co-immunoprecipitated by paxillin antibodies (±SEM, n = 3; *p<0.05, *t* test).

*Figure 4 continued on next page*

*Figure 4 continued*

DOI: https://doi.org/10.7554/eLife.31101.012

The following figure supplement is available for figure 4:

**Figure supplement 1.** Paxillin associates with the F-BAR domain of CIP4.

DOI: https://doi.org/10.7554/eLife.31101.013

and the last two LD domains (as seen in the paxillin<sup>LD1-3</sup> construct) was required to completely attenuate binding of vinculin and CIP4 to paxillin (see *Figure 4C,D*). Given that CIP4 dimerizes in vitro and in vivo via its F-BAR domain (as verified by small angle X-ray scattering analysis), we asked whether increased CIP4 binding to paxillin LIM domains may spatially hinder vinculin from accessing the LD motifs. Indeed, ectopic expression of full-length CIP4 or the F-BAR domain alone in HEK293T cells decreased association of both GST-PXN<sup>FL</sup> (*Figure 4E*) and endogenous paxillin (*Figure 4F*) with vinculin dose-dependently, but ectopic expression of the F-BAR-deleted CIP4 mutant (ΔN'-F-BAR) did not (*Figure 4F*, and *Figure 4—figure supplement 1B–D*). In addition, competition assays using the paxillin variants GST-PXN<sup>FL</sup>, GST-PXN<sup>ΔLD1</sup>, and GST-PXN<sup>ΔLIM3-4</sup> showed that the CIP4-binding LIM 3–4 region, but not the vinculin-binding LD1 motif, is required for high affinity CIP4 binding, as CIP4 could not out-compete vinculin for binding to the paxillin-ΔLIM3-4 construct (*Figure 4E*, and *Figure 4—figure supplement 1F*). These findings indicate that endocytic factors interact with the paxillin LIM domain and are suggestive of a competitive-binding mechanism.

## Paxillin is required for robust neurite formation on soft substrates

Given that endocytic activity is required for neurite initiation on a soft substrate, we wondered whether paxillin and CIP4 function in endocytosis. To assess this, we evaluated FM4-64 uptake in hippocampal neurons grown on soft substrates with or without paxillin/CIP4 knockdown by shRNA (*Figure 5—figure supplement 1*). Reducing paxillin and CIP4 levels in this way significantly decreased the rate and level of FM4-64 uptake (endocytosis) in neurons grown on soft gels (0.1 kPa; *Figure 5A,C* and *Figure 5—figure supplement 2*). This outcome was not seen following ectopic expression of the shRNA-resistant paxillin mutant PXN-R (*Figure 5A,C*). Furthermore, ectopic expression of full-length paxillin potentiated FM4-64 uptake in neurons grown on both soft and stiff gels (0.1 kPa or 20 kPa; relative to untransfected neighboring control cells); an effect not seen following ectopic expression of paxillin<sup>ΔLIM3-4</sup> or paxillin<sup>LD1-3</sup>, which lacks the LIM-domain required for CIP4 association (*Figure 5B,D and E*). These findings suggest that paxillin assists endocytosis on soft substrates via its interaction with components of the endocytic machinery. Note that in neurons cultured on 20 kPa gels, but not on 0.1 kPa gels, we observed an altered CIP4 distribution along the enlarged lamella edge (*Figure 5—figure supplement 2*); a pattern similar to that reported for coverslips (*Saengsawang et al., 2012*) on which the endocytic role of CIP4 may have been compromised.

Based on its physical and functional association with CIP4 and CHC (*Figure 4* and *Figure 5*), we wondered whether paxillin might participate in vesicle invagination during clathrin-coated pit formation. To test this possibility, we quantified QD-BDNF invagination in HEK293T cells with or without paxillin knockdown. Note that HEK293T cells respond to BDNF only when transfected with a BDNF receptor (TrkB) expression construct (see *Figure 6A*). We did not observe detectable QD-BDNF invagination into non-TrkB-expressing HEK293T cells but, following transfection of cells with a TrkB expression construct, we observed an average invagination rate of ~32 dots per cell over the first hour after bath application of QD-BDNF. That rate significantly decreased to four dots per cell on paxillin knockdown, and comparable results were seen following CIP4 knockdown or dynasore treatment (*Figure 6A,B*). The decreased rates were rescued by ectopic expression of shRNA-resistant PXN-R in the paxillin knockdown experiment (*Figure 6A*). These findings suggest that paxillin is required for efficient vesicle invagination.

The effect of paxillin loss on neurite formation was further evaluated by measuring the rate at which paxillin knockdown cells enter a segmentable state. On soft substrates (0.1 kPa), paxillin knockdown significantly increased the percentage of neurons exhibiting the BL morphology and reduced the number of neurons bearing neurites in 16 hr cultures, with these effects being similar to those seen following treatment with the endocytosis inhibitors (*Figure 2B*, and *Figure 7A,B*). Further, paxillin knockdown significantly reduced levels of active Rac1 accumulation at the protruding

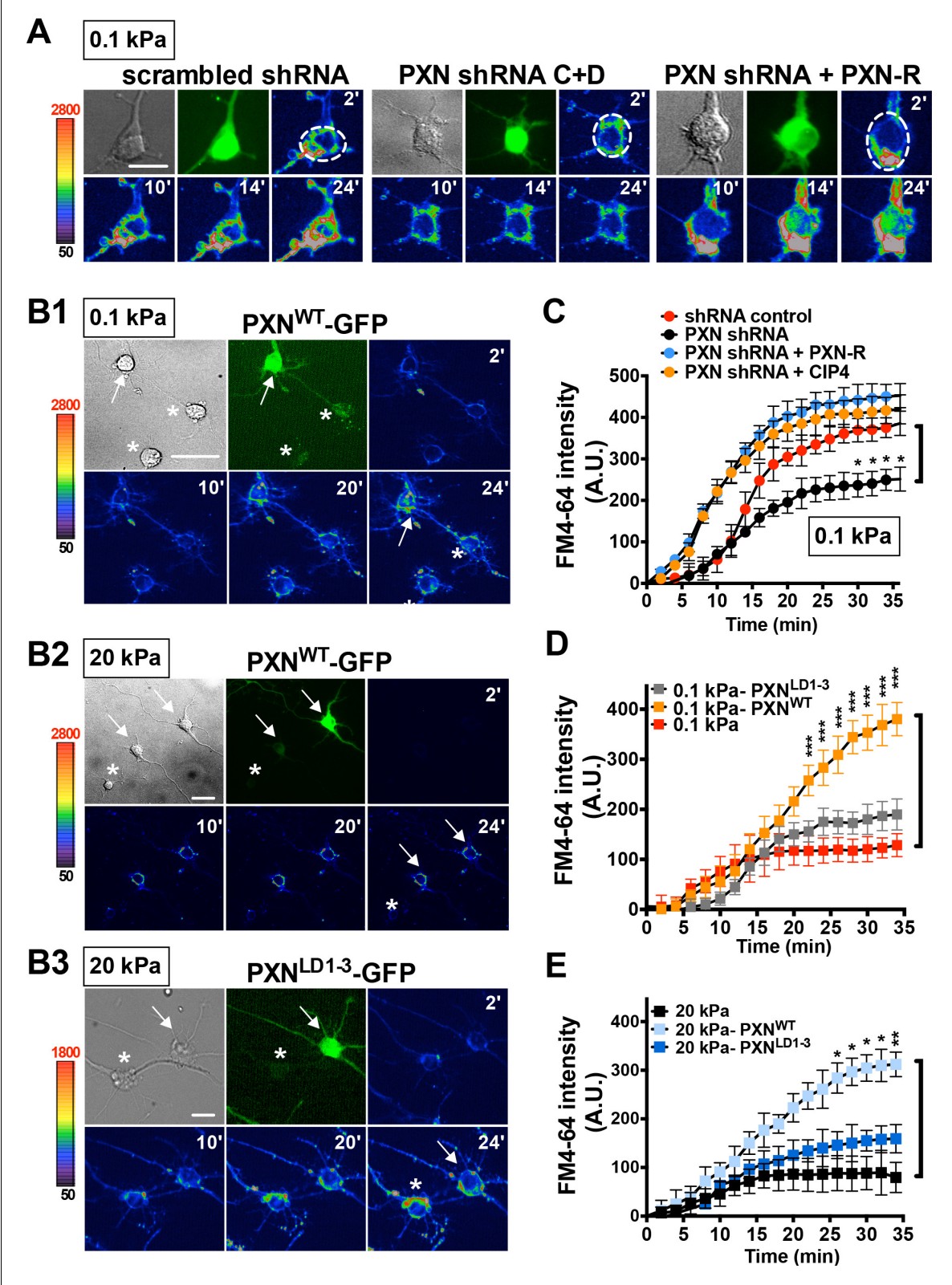

**Figure 5.** Paxillin is required for endocytosis promoted by a soft surface. (A) Paxillin knockdown suppresses the endocytic activity of neurons grown on 0.1 kPa gels. Representative time-lapse images of FM4-64 uptake in 3-DIV neurons on substrates of varying elasticity. Hippocampal neurons on 0.1 kPa gels were transduced with lentiviral particles harboring an shRNA-resistant construct ('PXN-R') and/or constructs harboring scrambled control or paxillin ('PXN shRNA C + D') shRNA at 5 hr after cell plating. Dashed circles surround the region of interest (ROI) in quantitative FM4-64 measurements. Bar: 20

*Figure 5 continued on next page*

*Figure 5 continued*

µm. (B) Similar to A, except constructs encoding GFP fusions of wild-type paxillin ('PXN^WT-GFP'; **B1 and B2**) or the corresponding LIM domain deletion mutant ('PXN^LD1-3-GFP'; **B3**) were used for lentiviral transduction. Asterisk: non-transduced neighboring cells. Arrows: neurons expressing GFP-tagged paxillin proteins. Bar: 20 µm. (C–E) Quantitative measurements of cumulative FM4-64 intensity (±SEM, n > 3 independent experiments, 7–12 cells for each set of experiments; *p<0.05; **p<0.001; ***p<0.0001; compared to control groups; multiple *t* tests), all from experiments similar to those described in A and B. Note that ectopic expression of wild-type paxillin, but not PXN^LD1-3, restored rapid endocytic FM4-64 uptake on 20 kPa stiff gels.
DOI: https://doi.org/10.7554/eLife.31101.014

The following figure supplements are available for figure 5:

**Figure supplement 1.** Knockdown efficiency of PXN shRNAs.
DOI: https://doi.org/10.7554/eLife.31101.015

**Figure supplement 2.** Endocytic function and distribution patterns of CIP4 protein in 0.1 kPa neuronal cultures.
DOI: https://doi.org/10.7554/eLife.31101.016

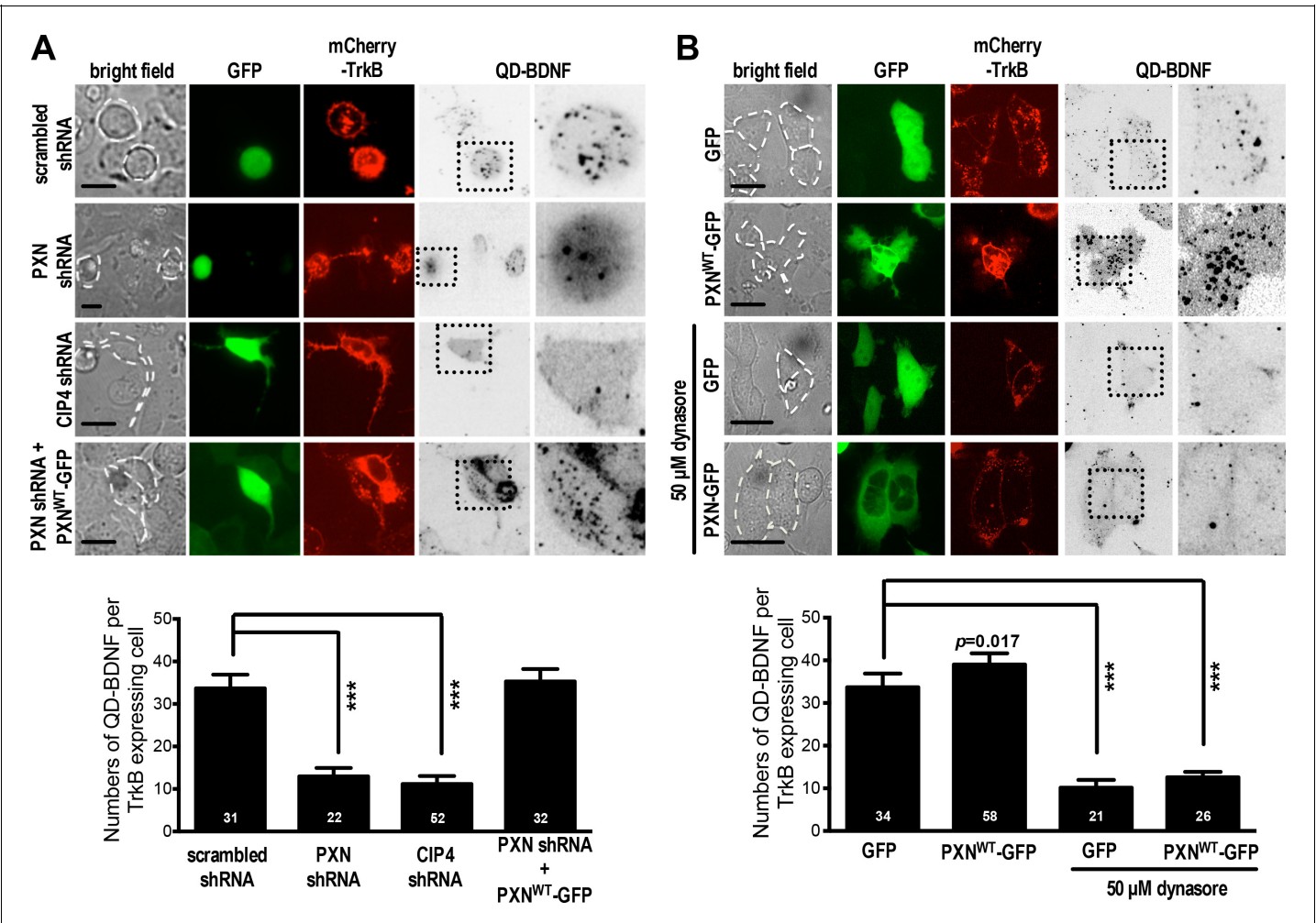

**Figure 6.** Paxillin facilitates QD-BDNF uptake. (A) QD-BDNF internalization assay on HEK293T cells co-transfected with expression vectors encoding the BDNF receptor TrkB (Red), together with shRNA constructs (Green) and/or wild-type paxillin, as indicated. Histograms showing that co-transfection of paxillin shRNA C and shRNA D significantly reduced QD-BDNF internalization (as reflected by an ~8-fold reduction in the quantity of intracellular QD-BDNF). Data represent means ±SEM (n ≥ 3 independent experiments, 26–52 cells for each set of experiments; compared to scrambled shRNA control; ***p<0.001; ANOVA with *Dunnett's post hoc* test). (B) Similar to A, except cells were pre-incubated with or without the dynamin inhibitor dynasore (50 µM, 30 min). Histograms show that dynamin is required for paxillin to promote QD-BDNF internalization. Data represent means ±SEM (n ≥ 3 independent experiments, 21–58 cells for each set of experiments; compared to GFP control; ***p<0.001; ANOVA with *Dunnett's post hoc* test).
DOI: https://doi.org/10.7554/eLife.31101.017

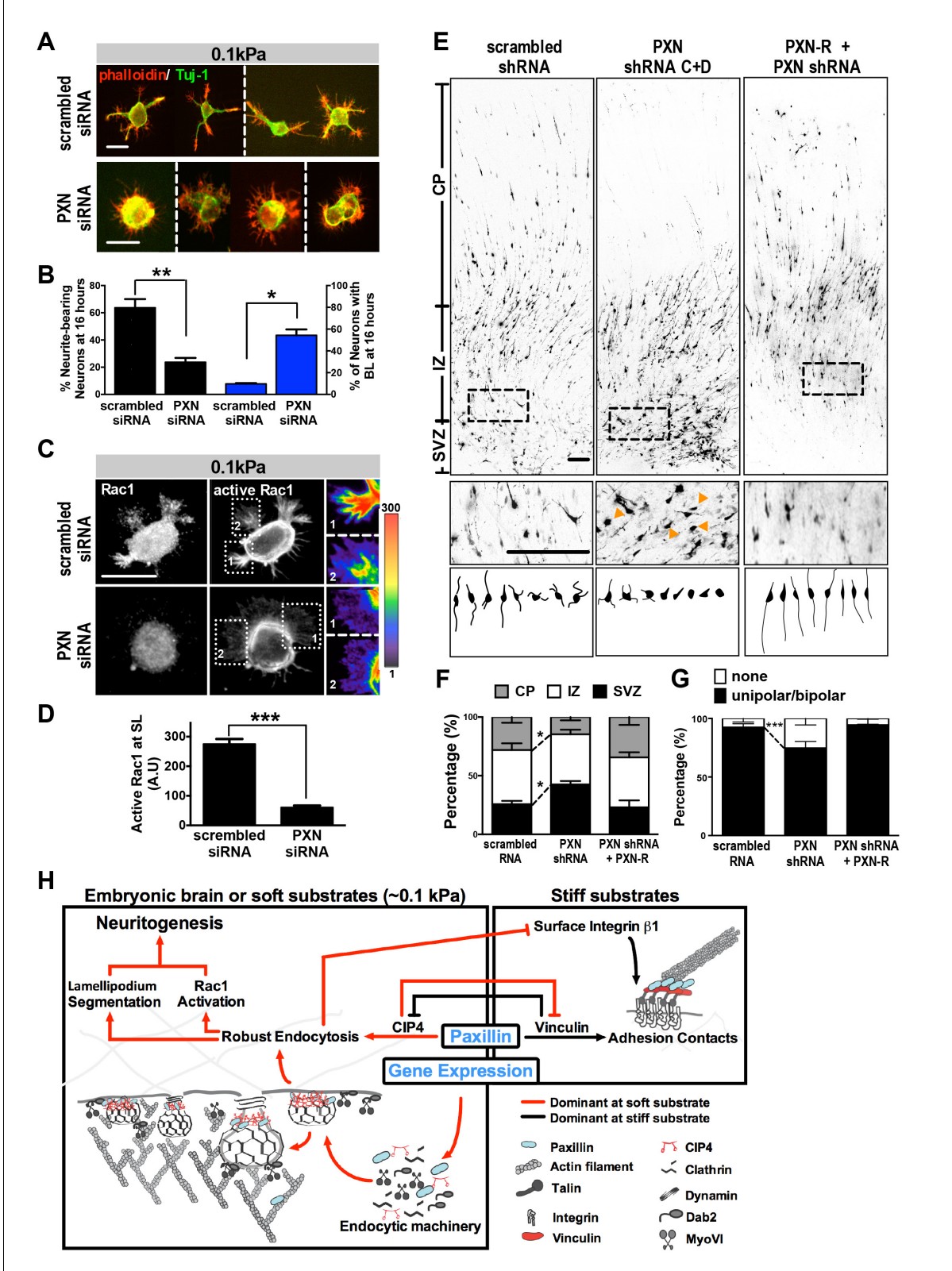

**Figure 7.** Paxillin knockdown alters neurite formation and neuronal migration in vitro and in vivo. (A and B). Paxillin knockdown impairs lamellipodium segmentation and neurite formation in neuronal cultures on 0.1 kPa gels. (A) Images of 16 hr hippocampal neurons transfected with scrambled control or paxillin siRNA at 30 min after cell plating on 0.1 kPa gels, followed by phalloidin staining for F-actin (Red) and immunostaining for the neuronal marker Tuj-1 (Green) at 16 hr after cell plating. (B) Histograms showing that paxillin knockdown significantly decreased the percentage of neurite-

*Figure 7 continued on next page*

*Figure 7 continued*

bearing neurons (left y-axis) and increased the percentage of neurons exhibiting the BL phenotype (right y-axis) in 0.1 kPa cultures. Data represent mean (±SEM; n > 3 independent experiments, >250 cells for each group; *p<0.05; **p<0.01; t test). (**C**) Similar to **A**, except that 16 hr neuronal cultures were immunostained with antibodies against active Rac1-GTP (Gray). Fluorescence intensity of active Rac1 (boxed region) is coded by pseudocolors in the linear scale (right panel in **C**). Scale bar: 20 μm. (**D**) Quantification of active Rac1 levels at segmented lamellipodia (averaged pixel value of a 2 μm X 2 μm area) in 0.1 kPa cultures transfected with paxillin-siRNA or control scrambled siRNA. Data represent mean ±SEM (n > 3 independent experiments, >250 cells for each group; ***p<0.001; ANOVA with *Dunnett's post hoc* test). (**E**) Paxillin knockdown promotes aberrant neurite formation and cortical neuron migration in vivo. Fluorescence images of E20 rat cortices transfected in utero at E17.5 with IRES constructs harboring EGFP plus scrambled-shRNA control, paxillin shRNA C + D, and/or shRNA C + D resistant PXN-R. The middle panels show 4x magnifications of boxed regions of the corresponding E20 cortex in the top panels. Bottom panels show sample tracings of 2D projections from confocal images of eight typical cortical neurons in the subventricular zone ('SVZ') of the corresponding E20 cortex. Bar, 100 μm. (**F**) Histograms showing the percentage (±SEM, n = 3 cortices each; *p<0.05, two-tailed *t* test) of neurons residing in the cortical plate ('CP'), intermediate zone ('IZ'), or subventricular zone ('SVZ') regions. (**G**) Calculation of the percentages (±SEM, n > 150 cells per cortex,>5 cortices each; ***p<0.001, multiple *t* test) of transfected cortical neurons exhibiting unipolar/bipolar processes or no process ('none'; arrowheads in e) in the cortical SVZ/IZ region for cortices described in **e**. Datasets (connected by dashed lines) showing significant differences are marked. (**H**) Schematic illustrating the proposed substrate elasticity-controlled, paxillin-dependent bistable mechanism, comprising a genetic response and mutual inhibition of endocytosis by adhesion and vice versa. Growth on a soft substrate shifts neurons to a Rac1-activated neuritogenic state.

DOI: https://doi.org/10.7554/eLife.31101.018

The following figure supplements are available for figure 7:

**Figure supplement 1.** Manipulation of paxillin-binding affinity leads to a SL-BL phenotypic switch.
DOI: https://doi.org/10.7554/eLife.31101.019

**Figure supplement 2.** Differential protein abundances between the endocytosis and adhesion machineries in embryonic brain, cardiac, and hepatic tissues.
DOI: https://doi.org/10.7554/eLife.31101.020

**Figure supplement 3.** Expression time-course of adhesion- or endocytosis-related proteins in developing mouse cerebral cortex.
DOI: https://doi.org/10.7554/eLife.31101.021

edge (*Figure 7C,D*). These findings suggest that paxillin is a putative factor participating in a substrate modulus-dependent bistable switch in neurite formation in vitro (*Figure 7H*). We also asked whether manipulating paxillin binding affinity for endocytic or adhesion molecules could lead to a SL-BL phenotypic switch. Based on the domain mapping results in *Figure 4C*, we conclude that the paxillin-ΔLD1 variant favors CIP4/clathrin association and that paxillin-ΔLIM3-4 favors vinculin association. Using this approach, we found that ectopic PXN$^{\Delta LIM3-4}$ expression in 0.1 kPa cultures enhanced the neuritogenesis-delaying BL phenotype, whereas PXN$^{\Delta LD1}$ expression reduced the BL population in 20 kPa cultures (*Figure 7—figure supplement 1*). By contrast, ectopic expression of PXN$^{LD1-3}$, which cannot bind vinculin or CIP4, had no significant effect on neurite outgrowth (*Figure 7—figure supplement 1*). These results support the idea that whether paxillin forms a complex with either endocytic or adhesion molecules determines the probability that newborn neurons will form proper neurites.

## Paxillin is required for neurite development in vivo

We next evaluated the effect of paxillin activity on neurite development in vivo by monitoring the protein expression of endocytic factors in embryonic rat brain. We observed high levels of endocytosis-related factors and low levels of adhesion-related molecules in embryonic rat brains compared to hepatic and cardiac control tissues (*Figure 7—figure supplement 2*). In addition, paxillin abundance increased between E13 and E18 and declined postnatally (between P1 and P14) in the developing cortex (*Figure 7—figure supplement 3*). This time-course is compatible with and parallels that of neurite formation in the developing rat brain.

Then, to assess the function of paxillin in neurite development in vivo, we conducted in utero electroporation of paxillin shRNA (PXN-shRNAs) in a subpopulation of neural progenitor cells at embryonic day 17 (E17). Because aberrant neurite formation in vivo usually delays neuronal migration into the cortical plate (CP) (*Hsu et al., 2015*), we analyzed E20 rat brain slices in which most cells (~75%) in control embryos had migrated out of the subventricular zone (SVZ) with a radial infiltration into the intermediate zone (IZ) toward the CP (*Figure 7E,F*). Neurons expressing paxillin-shRNA showed apparent migration defects relative to controls, with ~42% of cells accumulating in

the SVZ, wherein ~25% of paxillin-knockdown neurons did not exhibit neurite processes (*Figure 7E, G*). These phenotypes were rescued by co-expression of shRNA-resistant PXN-R (*Figure 7E–G*).

## Discussion

Neurite formation is the first morphogenetic step in establishing axonal projections and dendritic territory. Under physiological environments in vivo, both genetic programs and cell mechanics contribute to timely emergence of neuronal morphology and activity. Nevertheless, it remains challenging to define spatio-temporal patterns of neurite initiation and most studies to date have tackled this question using thin-glass or plastic cell culture substrates in which cell-matrix adhesion is a prominent response. In the present study, we used culture on soft substrates to demonstrate a paxillin-linked bistable switch that governs the ability of newborn neurons to shift their lamellipodium morphology to one of two stable states and to assume a neuritogenic state within a few hours. Our system recapitulates temporal and morphologic features observed in vivo. By contrast, when cells are grown on rigid substrates or when endocytosis is inhibited, neurons adopt the behavior of most migratory cells and paxillin becomes associated with cell-matrix or adhesion factors, exhibiting broad lamellipodia that delay neurite formation. Such biphasic neurite initiation emerges from an amplification loop enabling mutual inhibition of endocytosis by the adhesion machinery, and vice versa (*Figure 7H*). Through this mechanism, newborn neurons can respond to the soft environments that dictate a phenotypic switch to allow neurite initiation. Our findings also complement previous views of neurite formation defined by conventional Banker's culture on coverslips.

To form protrusions from a cell body, cells must coordinate several distinct intracellular mechanisms, including actin retrograde flow (*Endo et al., 2003*; *Flynn et al., 2012*), actomyosin-generated contractility (*Amano et al., 1998*; *Raucher and Sheetz, 2000*), membrane fusion and/or membrane tension (*Fujita et al., 2013*; *Zheng et al., 1991*), and exo- and endocytic pathways (*Raiborg et al., 2015*). It is not surprising that paxillin is involved in these processes, given its role as an integrator of integrin and growth factor signaling (*Brown and Turner, 2004*). What is striking is our finding that paxillin plays a dual role in adhesion and endocytosis, with predominance of one of these roles determining the fate of a newborn neuron. Historically, paxillin has been known as a major player in cell-matrix adhesions, where it associates with tyrosine kinases (such as FAK and Src) and the actin-binding proteins vinculin and actopaxin (*Brown et al., 1996*; *Brown and Turner, 2004*; *Nikolopoulos and Turner, 2000*; *Schaller, 2001*; *Turner, 2000*). Our findings suggest that on soft gels in particular, newborn neurons recruit paxillin to the endocytic machinery (via binding with clathrin and CIP4) with a higher, but not exclusive, preference over cell-matrix adhesions. Our microarray analysis indicates that selectively elevated levels of endocytic factors may underlie their preferential recruitments with paxillin observed in soft-gel cultures. This heretofore unknown behavior of paxillin then elevates endocytic activity, Rac1 signaling, and expression of the genes encoding endocytic proteins that are required for timely neurite emergence. As a result, a soft environment with elasticity lower than 1 kPa is sufficient to induce timely neurite initiation. Conversely, our observations suggest that culturing cells on rigid substrates, disruption of Rac1 activation, and blockage of endocytosis all delay spontaneous neurite initiation.

Proper functioning of neural networks depends on physical connections between neurons, such as synapses, which in turn requires the capacity to control morphological changes such as protrusion in the course of neuritogenesis. For most cells, microenvironmental stiffness has been shown to promote cell spread and protrusion (*Discher et al., 2005*; *Yeung et al., 2005*). However, those same environments may lead to formation of unnecessary connections in young neurons or even promote deleterious 'noise' that can interfere with neuronal function. The ability of neonatal neurons to delay neuritogenesis on a rigid environment could endow cells with the ability to establish robust and functional networks only in permissive environments (i.e. soft microenvironments). While paxillin functions in endocytic reactions as opposed to adhesion upon neurite initiation on soft substrates, endocytic accessory components could have an unconventional role when endocytic signaling becomes compromised under high cell-matrix adhesion regimes. For example, ectopically expressed CIP4 has been shown to accumulate at broad lamellipodia and to inhibit neurite formation on stiff substrata (coverslips) (*Saengsawang et al., 2012*), where CIP4 may function in actin polymerization rather than neuritogentic endosomal reactions.

Paxillin-associated endocytosis may also occur in epithelial cells, since both epithelial cells and neurons are derived from ectoderm. In fact, in vitro experiments demonstrate that proper development of epithelial organs requires soft microenvironments, while stiff stroma is often observed in advanced cancers (*Gilkes et al., 2014*; *Paszek et al., 2005*; *Wirtz et al., 2011*). By comparison, muscle cells, fibroblasts or osteoclasts, which are derived from other germ layers, may exhibit higher tension or rigidity. Based on studies reporting the natural stiffness of mouse embryonic cerebral cortex at E16.5 to E18.5 (*Iwashita et al., 2014*), we surmise that a neuronal switch favoring paxillin/ endocytic factor binding is required and activated immediately after embryonic neuronal differentiation to ensure timely neurite initiation. This switch may also promote departure of cells from neurogenic regions, such as the ventricular and subventricular zones, whose stiffness is relatively high compared to the cortical plate at E18.5. However, at a later developmental time-point (since brain stiffness increases with age) and at synaptic contact sites, the paxillin adhesion switch may predominate, allowing contacts with postsynaptic cells or decreasing the capacity for neurite outgrowth. For example, synapses at neuro-muscular junctions where repetitive muscle contraction can endow much greater localized stiffness than that seen in brain cortex may favor paxillin/adhesion machinery association. Furthermore, as shown here, the paxillin adhesion switch relies on formation of adhesion complexes and consequently depends on integrin signaling. In developing and adult brain, integrins play important roles in controlling neuronal process outgrowth and regulating synaptic plasticity and memory formation (reviewed in [*Park and Goda, 2016*]). However, the paxillin-mediated switch to control timing of neurite initiation may not underlie morphogenetic events occurring at later stages, as neuronal paxillin expression decreases as neurons mature. Therefore, it will be important to revisit the relationship among genetic profiles, cellular reactions, and tissue mechanical properties over different cell types and developmental stages.

Looking forward, our observations and our model for a molecular switch that is integrated with changes in gene expression and the dual role of paxillin represent a robust mechanism that could underlie synchronous behaviors, such as morphological transformation or cohort migration, that occur when a group of developing cells is exposed to diverse signals in vivo. Besides substrate stiffness, physiological constraints imposed by other factors, such as cell-cell contact guidance or actin-microtubule coupling efficiency, could play important roles in neurite formation. In addition to a morphogenetic role, paxillin-associated endocytosis may also potentiate retrograde relay of growth and survival signals deployed from the neuritic terminal (*Cosker and Segal, 2014*) and thereby provide another mechanism to positively regulate neuronal development.

## Materials and methods

**Key resources table**

| Reagent type (species) or resource | Designation | Source or reference | Identifiers | Additional information |
|---|---|---|---|---|
| cell line | HEK293T | ATCC | ATCC Cat# CRL-3216, RRID:CVCL_0063 | |
| antibody | anti-FLAG M2 | Sigma-Aldrich | Sigma-Aldrich Cat# F1804, RRID:AB_262044 | dilution: 1:1000 |
| | c-Myc Tag Monoclonal Antibody (9E10) | Thermo Fisher Scientific | Thermo Fisher Scientific Cat# MA1-980, RRID:AB_558470 | dilution: 1:1000 |
| | Anti-beta III tublin Antibody | Millipore | Millipore Cat# AB9354, RRID:AB_570918 | dilution: 1:1000 in IF |
| | MAP2 | Millpore; PMID:26244549 | Millipore Cat# AB5622, RRID:AB_11213363 | dilution: 1:1000 in IF |
| | actin | Millpore | Millipore Cat# MAB1501, RRID:AB_2223041 | dilution: 1:2000 |
| | Tau-1 | Millpore | Millipore Cat# MAB3420, RRID:AB_94855 | dilution: 1:1000 in IF |

*Continued on next page*

Continued

| Reagent type (species) or resource | Designation | Source or reference | Identifiers | Additional information |
|---|---|---|---|---|
| | Anti-Paxillin (N-Term), Rabbit Monoclonal, clone Y113 antibody | Millpore | Millipore Cat# 04–581, RRID:AB_838293 | dilution: 1:1000 in WB; 1:200 in IF |
| | p-PaxillinY118 | ECM Biosciences | ECM Biosciences Cat# PP4501 | dilution: 1:1000 |
| | Mouse Anti-Paxillin Monoclonal Antibody, Unconjugated, Clone 349 | BD Biosciences; PMID:28362576 | BD Biosciences Cat# 612405, RRID:AB_647289 | dilution: 1:1000 |
| | Anti-Vinculin, clone V284 antibody | millpore; PMID:28697342 | Millipore Cat# 05–386, RRID:AB_309711 | dilution: 1:1000 in WB; 1:200 in IF |
| | Anti-Integrin Beta1, activated, Clone HUTS-4, Azide Free antibody | Millore; PMID:28602620 | Millipore Cat# MAB2079Z, RRID:AB_2233964 | dilution: 1:1000 |
| | TRIP/CIP4 | Bethyl Laboratories; A301-186A | | dilution: 1:1000 in WB; 1:200 in IF |
| | Clathrin heavy chain antibody | Abcam; PMID:28231467, PMID:28575669 | Abcam Cat# ab21679, RRID:AB_2083165 | dilution: 1:1000 in WB; 1:200 in IF |
| | Talin 1 and 2 antibody [8D4] | Abcam | Abcam Cat# ab11188, RRID:AB_297828 | dilution: 1:1000 in WB |
| | Rabbit Anti-Rab5 Polyclonal Antibody, Unconjugated | Abcam; PMID:28408870, PMID:28669519 | Abcam Cat# ab18211, RRID:AB_470264 | dilution: 1:1000 in WB |
| | Integrin beta 1 antibody [EP1041Y] - Carboxyterminal end | Abcam; PMID:25330147, PMID:28552668, PMID:28609658 | Abcam Cat# ab52971, RRID:AB_870695 | dilution: 1:1000 in WB; 1:200 in IF |
| | Phospho-FAK (Tyr397) Antibody (31H5L17), ABfinity(TM) Rabbit Monoclonal | Thermo Fisher Scientific; PMID:22049075, PMID:25280968, PMID:26056143, PMID:26381152, PMID:26393679, PMID:26984758 | Thermo Fisher Scientific Cat# 700255, RRID:AB_2532307 | dilution: 1:1000 in WB; 1:200 in IF |
| | Rabbit Anti-FAK [pY397] Polyclonal Antibodies, Unconjugated antibody | Thermo Fisher Scientific; PMID:27474796, PMID:28069919, PMID:28520937 | Thermo Fisher Scientific Cat# 44–624G, RRID:AB_2533701 | dilution: 1:1000 in WB; 1:200 in IF |
| | Akt1/2/3 (H-136) antibody | Santa Cruz Biotechnology; PMID:27316329, PMID:27410235, PMID:28911175 | Santa Cruz Biotechnology Cat# sc-8312, RRID:AB_671714 | dilution: 1:200 in WB |
| | Rac 1 (C-11) antibody | Santa Cruz Biotechnology | Santa Cruz Biotechnology Cat# sc-95, RRID:AB_2176125 | dilution: 1:200 in WB |
| | Cdc42 (P1) antibody | PMID:28181299, PMID:28457749 | Santa Cruz Biotechnology Cat# sc-87, RRID:AB_631213 | dilution: 1:200 in WB |
| | Rho A (26C4) antibody | Santa Cruz Biotechnology; PMID:28287395, PMID:28323616 | Santa Cruz Biotechnology Cat# sc-418, RRID:AB_628218 | dilution: 1:200 in WB |
| | Anti-Active Rac1-GTP Mouse Monoclonal Antibody | NewEast Biosciences | NewEast Biosciences Cat# 26903, RRID:AB_1961793 | dilution: 1:200 in WB and IF |
| | Anti-RhoA antibody (mouse MAb)+control | Cytoskeleton Inc; PMID:27822498 | Cytoskeleton Cat# ARH03, RRID:AB_10708069 | dilution: 1:200 in WB and IF |
| | Anti-Rac1 specific mouse MAb antibody | Cytoskeleton Inc | Cytoskeleton Cat# ARC03-A, RRID:AB_10709099 | dilution: 1:200 in WB and IF |
| | Anti-Cdc42 Mouse Monoclonal Antibody | Cytoskeleton Inc | Cytoskeleton Cat# ACD03-A, RRID:AB_10716593 | dilution: 1:200 in WB and IF |
| peptide, recombinant protein | PAK-GST, GST-Photekin-RBD | Cytoskeleton Inc | Cytoskeleton Inc Cat. #BK035; Cat. # BK036 | |

## Plasmids

Plasmids used are as follows: fluorescence resonance energy transfer (FRET) reporters for Rac1 activity assays, Rac1-2G (Plasmid #66111; [Fritz et al., 2013], GFP-Dynamin 2 K44A (Plasmid #22301; (Ochoa et al., 2000) and paxillin-pEGFP plasmid (Plasmid # 15233; (Laukaitis et al., 2001) were obtained from Addgene. Lentiviral transfer plasmid pLenti-C-Myc-DDK-CIP4 (also known as TRIP10) was generated by subcloning the C-Myc-DDK-CIP4 fragment from pCMV-C-Myc-DDK-TRIP10 (OriGene Technologies, Rockville, MD) into pLenti vector using AsiSI and PmeI restriction sites. GST- or His-tagged fusion protein expression constructs encoding full-length or truncated versions of paxillin or CIP4 were cloned into pDONR/Zeo vectors (Invitrogen, Carlsbad, CA) by PCR-based methods and transferred to T7 promoter-driven pDEST15 or pEXP2-6xHis-DEST Gateway destination vectors (Invitrogen, Carlsbad, CA) by recombination-based Gateway technology according to the manufacturer's protocols (Invitrogen, Carlsbad, CA). Paxillin-mCherry plasmid was generated by replacing the EGFP fragment from paxillin-pEGFP with the mCherry sequence. Lentiviral-based short hairpin RNA (shRNA) constructs were purchased from OriGene Technologies (Rockville, MD). For paxillin shRNA constructs, 29-oligonucleotide duplexes targeting rat paxillin cDNA sequences (PXN-shRNA A targeting position 1505, PXN-shRNA B targeting position 1571, PXN-shRNA C targeting position 346, and PXN-shRNA D targeting position 952 of the corresponding paxillin sequence) and a control non-effective scrambled shRNA cassette (5'- CACAAGCTGGAGTACAACTACAACAGCCA-3') were cloned into pGFP-C-shLenti vector (OriGene Technologies, Rockville, MD). For CIP4 shRNA constructs, TRIP10-shRNA A targeting position 172, TRIP10-shRNA B targeting position 351, TRIP10-shRNA C targeting position 811, and TRIP10-shRNA D targeting position 983 of the corresponding CIP4 coding sequence were also cloned into pGFP-C-shLenti vector (OriGene Technologies, Rockville, MD). The encoding regions of shRNAs contained a 29-nucleotide sense strand, a TCAAGAG loop structure, and the reverse complementary sequence, followed by a TTTTTT termination sequence. The chicken ortholog of a mammalian paxillin expression plasmid, paxillin-pEGFP, was used as the shRNA-resistant construct (PXN-shRNA-R), the sequence of which is resistant to PXN-shRNA targeting. pLenti-C-mGFP-PXN (rat paxillin) was purchased from OriGene Technologies (Rockville, MD). The plasmid encoding mCherry-TrkB used in QD-BDNF internalization assays was kindly provided by Dr. Mu-Ming Poo, UC Berkeley, USA.

## Reagents

Sources of antibodies, proteins, and chemicals are as follows. Purified PAK-GST and GST-Rhotekin-RBD for active Rac1 and RhoA pulldown assays were from Cytoskeleton Inc. (Denver, CO). Primary antibodies: monoclonal anti-FLAG M2 was from Sigma-Aldrich (Saint Louis, MO). Polyclonal antibodies to neuronal class III β-tubulin (Tuj-1, AB9354) and MAP2 (AB5622), as well as monoclonal antibodies to actin (clone C4, MAB1501), Tau-1 (clone PC1C6, MAB3420), paxillin (N-term, 04–581), vinculin (clone V284, 05–386) and activated integrin β1 (clone HUTS-4, MAB 2079Z) were from EMD Millipore Corp. (Billerica, MA). Polyclonal antibodies to TRIP/CIP4 (A301-186A). was from Bethyl Laboratories, Inc. (Montgomery, TX). Clathrin heavy chain (ab21679), Rab5 (ab18211), and monoclonal antibodies to integrin β1 (ab52971) and talin 1/2 (ab11188) were from Abcam Inc. (Cambridge, MA). Polyclonal antibody to p-FAK$^{Y397}$(44–624G) and monoclonal antibody to p-FAK$^{Y397}$ (700255) were from Thermo Fisher Scientific. Polyclonal antibody to p-Paxillin$^{Y118}$ (PP4501) was from ECM Biosciences (Versailles, KY). Monoclonal antibody to paxillin (612405) was from BD Biosciences (Franklin Lakes, NJ). Polyclonal antibodies to Akt1/2/3 (H-136), Rac1 (C-11, sc-95), Cdc42 (P1, sc-87), as well as monoclonal antibody to RhoA (26C4, sc-418), were from Santa Cruz Biotechnology (Santa Cruz, CA). Monoclonal antibody to c-Myc (9E10, MA1-980) was from Thermo Scientific (Waltham, MA). Monoclonal antibody to active Rac1-GTP (26903) was from NewEast Biosciences (King of Prussia, PA). Monoclonal antibodies to RhoA (ARH03), Rac1 (ARC03) and Cdc42 (ACD03) were from Cytoskeleton Inc. (Denver, CO).

Pharmacological reagents: recombinant human BDNF was from PeproTech (Rocky Hill, NJ). PP2 (Src and RIP2 kinase inhibitor, ab120308), Dynole2–24 (potent dynamin I and II inhibitor; ab141290), and Dyngo 4a (ab120689; highly potent dynamin inhibitor) were purchased from Abcam Inc. (Cambridge, MA). Actinomycin D (A9415), dynasore (D7693) and dansylcadaverine (also described as monodansyl cadaverine (MDC), 30432) were purchased from Sigma Aldrich (Saint Louis, MO).

PD150606 (1269) and NSC23766 (2161) were purchased from Tocris Bioscience (Ellisville, MO). LY 294002 (440202) and Y-27632 (SCM075) were bought from EMD Millipore Corp. (Billerica, MA).

Fluorescent reagents: Alexa Fluor 488, Alexa Fluor 546, Alexa Fluor 647, fluorescent dye FM1-43FX (F35355) and FM4-64FX (F34653) were from Invitrogen (Carlsbad, CA). DAPI (D9542), Acrylamide (A4058) and N'-N'-methylenebisacrylamide solution (M1533) were purchased from Sigma-Aldrich (Saint Louis, MO). Phalloidin-iflour 488 conjugate, phalloidin-iflour 555 conjugate and phalloidin-iflour 647 conjugate were from AAT Bioquest Inc. (Sunnyvale, CA).

## Cell culture, protein lysate preparations, and immunostaining

Hippocampal neurons were prepared from E17.5 rat embryos as previously described (*Dotti et al., 1988*), and were cultured in neurobasal medium supplemented with Gem21 NeuroPlex (GEMINI bio-products, West Sacramento, CA). A similar procedure was applied to preparation of cortical neuronal cultures. Human Embryonic Kidney 293T (HEK293T; ATCC Cat# CRL-3216) cells used for biochemical assays were tested for Mycoplasma and characterized by STR profiling as indicated in the ATCC online catalog. HEK293T cells were cultured in Dulbecco's Modified Eagle's Medium supplemented with 10% fetal bovine serum (Biological Industries, Beit Haemek, Israel). Transfection of these cultures was performed using a lentivirus-based expression system or Lipofectamine 2000 (Invitrogen, Carlsbad, CA), according to the manufacturer's instructions. Unless otherwise stated, hippocampal neurons were used as a standard model for in vitro immunocytochemistry to analyze neuronal morphology. Cortical neuronal cultures were used to obtain a sufficient number of cells for biochemical assays not requiring transfection of exogenous proteins.

For cortical neurons grown on gel or glass substrates, protein lysate was harvested in RIPA buffer (Sigma Aldrich, St. Louis, MO) containing complete protease inhibitor cocktail (Roche, BASEL, Switzerland) and phosphate inhibitor PhosSTOP (Roche, BASEL, Switzerland). For tissue lysate preparations, brain and heart were dissected out from E17.5 rat embryos and rinsed briefly with PBS, followed by tissue homogenization using a pestle in ice-cold lysis buffer (20 mM Tris-HCl, 5 mM $MgCl_2$ and 1 mM DTT), followed by four rounds of sonication (20 s on followed by 10 s off). For other biochemical experiments, cell lysates were prepared by the freeze-thaw method.

For immunostaining, cultured hippocampal neurons were fixed with 4% paraformaldehyde for 12 min and then permeabilized in 0.3% Triton X-100 for 12 min and blocked with 3% BSA for 1 hr. The fixed cells were processed further for immunostaining according to standard procedures and imaged with a confocal microscope (Zeiss LSM700) equipped with a 63 × oil immersion objective (NA1.4; Zeiss) and a 40 × water immersion objective (NA1.1; Zeiss). For quantitative measurements of colocalization, images were acquired using a Zeiss LSM880 confocal microscope with an Airyscan module (32-channel GaAsP detector array) equipped with a Plan Apo 63 × oil immersion objective (NA1.4). The axial step size was set to 125 nm. Surface rendering of the three-dimensional images was performed using ZEN (Zeiss) and Imaris (Bitplane) software without Z-correction. Images were analyzed and processed for presentation in the figures, using brightness and contrast adjustments with NIH ImageJ software and following the guidelines of Rossner and Yamada (*Rossner and Yamada, 2004*).

## Polyacrylamide gel preparation

Fabrication of polyacrylamide gels with tunable mechanical stiffness was slightly modified from a previous protocol (*Tse and Engler, 2010*). Briefly, uniform polyacrylamide gels were fabricated in a three-layer assembly of 200 μm thickness. The bottom layer was a hydrophilic amino-silanized coverslip prepared according to the following protocol. A thin film of sodium hydroxide was allowed to form on the coverslips at approximately 90°C. The entire surface of each coverslip was then immersed with a sufficient volume of (3-aminopropyl) triethoxysilane (Sigma-Aldrich, Saint Louis, MO) for 5 min. The (3-aminopropyl) triethoxysilane was then completely rinsed off to prevent precipitation before 0.5% (v/v) glutaraldehyde (Sigma-Aldrich, Saint Louis, MO) in PBS was added onto the silanized coverslips for 30 min. Fluids were removed by suction before the amino-silanized coverslips were air-dried and sterilized with 70% ethanol for 16 hr prior to gel preparation. The top layer comprised a laminin-coated coverslip prepared by sterilizing an acid-washed coverslip with 70% ethanol and then coating it with 5 μg/ml of poly-L-Lysine (Sigma Aldrich, Saint Louis, MO) and 0.08 μg/ml of laminin (Corning, NY) by absorption for an hour at 37°C.

For the middle gel layers of differing stiffness, a prepolymer mixture was prepared as shown in the table below. Polymerization of the prepolymer mixture was carried out by adding 10 µl of 10% ammonium persulfate, 2 µl of TEMED (Bio-Rad Laboratories, Hercules, CA) and sufficient deionized water to yield a final volume of 1000 µl. The resulting prepolymer-catalyst mixture was dropped onto hydrophobic amino-silanized coverslips. The three-layer assembly was formed by transferring the pre-coated poly-L-Lysine and laminin coverslips to the surfaces of the polyacrylamide gels during polymerization. After 30 min, the top layer was gently peeled off and washed three times with HEPES solution to remove unreacted monomer and excess coatings.

|  | 4% acrylamide (µl) | 2%N'-N'-methylenebisacrylamide (µl) | Deionized water (µl) |
| --- | --- | --- | --- |
| 0.1 kPa | 75 | 25 | 900 |
| 1 kPa | 100 | 50 | 850 |
| 20 kPa | 200 | 240 | 560 |

## Substrate elasticity measurements

Substrate elasticity of uniform gels was measured by using a JPK NanoWizard II AFM system installed above the stage of an inverted light microscope (Zeiss Axio Observer) in a custom-built anti-noise, anti-vibration system. A 5 µm (in diameter) polystyrene bead-modified tip-less cantilever (ARROW-TL1-50, NanoWorld, US) was utilized. The spring constants, calibrated by a thermal noise-based method, were at a range of 0.02 to 0.08 N/m for all cantilevers. All indentation depth curves were calculated using the manufacturer's software (Hertz model, JPK instruments, Berlin, Germany). The average value of Young's modulus of tensile elasticity was acquired from 25 measurements for each independent experiment.

## Phalloidin intensity profiling

Images of 5 hr neuronal cultures stained with phalloidin were analyzed in a circular coordinate system using an oval-profile ImageJ plugin (courtesy of Bill O'Connell, http://rsbweb.nih.gov/ij/plugins/oval-profile.html). The pixel intensities along each radian in a circle (radius = 8 µm) covering the cell periphery were normalized against the background value and plotted as an intensity-distribution profile. The lamellipodium occupancy (LO) refers to the proportion of the individual phalloidin arc length (angular span) to the entire cell periphery ($2\pi r$), calculated by the formula $\frac{\text{phalloidin arc length}}{2\pi r}$. The major LO represents the maximum LO value of each cell.

## Single molecule FISH

Neurons grown on gel or glass substrates were fixed in 4% paraformaldehyde at room temperature for 10 min, and then permeabilized in 0.3% Triton X-100 for 12 min and rinsed in 70% ethanol for subsequent RNA FISH. For hybridization, samples were briefly washed once with wash buffer (10% deionized formamide, 2 × SSC) for 5 min, and then hybridized with RNA FISH probes in hybridization buffer (10% formamide, 10% dextran sulfate, 2 × SSC) overnight at 37°C. Following hybridization, samples were washed twice with wash buffer (30 min per wash), and washed once with 1 × PBS. Samples were then imaged with a Delta Vision microscopy system equipped with a PlanApoN 60X oil-immersion objective (1.42 NA; Olympus). Images were collected and analyzed with NIH Image software. Alexa Fluor 488 phalloidin was used to identify the lamellipodium phenotypes of the stained cells. smRISH probe sets targeting the rat transcripts of *CLTC, MYO6, DAB2, VCL, CDH11, ROBO2, DKK1, and CYR61* were ordered from Stellaris with Quasar 570 dye (Bioserach Technologies, Petaluma, USA).

## FRET imaging and analysis

Cultured hippocampal neurons were imaged with a Rolera EM-C$^2$ EMCCD camera (QImaging) and Yokogawa CSU-X1 spinning-disk confocal microscopy (Zeiss) and a 40 × water immersion objective (NA1.1; Zeiss). Excitation spectra were excited by a solid-state 445 nm diode laser, through a 457 nm dichroic filter. Emission spectra were sequentially acquired using 485 ± 20 nm and 535 ± 30 nm band-pass emission filters for mTFP1 and Venus fluorescence, respectively. Images were collected

and analyzed with NIH ImageJ software. All filters and dichroics were from Chroma Technology. Live images were acquired for 150 milliseconds at 3 s intervals. The intensity of mTFP1 and Venus fluorescence was measured at a level below saturation for all neurites. Measurements were not performed on the soma due to fluorescence saturation being at the excitation level suitable for neurite measurements. For the ratiometric FRET analysis, the mTFP1 and Venus signals were background-subtracted (with background intensity taken from a nearby cell-free region), normalized against the control value (averaged over 3 min), and the FRET value was calculated as a ratio (mTFP1/Venus). The intensity of the FRET signal was calculated with NIH ImageJ software and is represented by pseudocolors.

## Expression of GST- and His-tagged proteins, and in vitro binding assays

For protein expression and purification, GST fusion proteins were produced as previously described. Briefly, GST- or His-tagged proteins were expressed in *E. coli* BL21(DE3) induced by 0.4 mM IPTG overnight at 30°C in LB medium. Bacteria were lysed by sonication in short pulses of 15 s in lysis buffer [(50 mM Tris-HCl, pH = 7.4, 50 mM NaCl, 5 mM DTT, 1 mM phenylmethylsulfonyl fluoride, 1% Triton X-100 containing 1% Triton X-100, 1 mM EDTA, 1 mM dithiothreitol and protease inhibitors (Complete EDTA-free; Roche)]. Cell debris was removed by centrifugation at 9000 *g* for 10 min at 4°C. The resulting supernatant was applied onto a Glutathione Sepharose 4B or HisTrap column (GE Healthcare). After washing, GST- and His-tagged proteins were eluted in buffer containing 10 mM reduced glutathione and 300 mM imidazole, respectively. We added 1 M imidazole to samples shortly after elution to prevent precipitation of His-tagged protein. Protein concentrations were measured using the Bio-Rad DC Protein Assay. Protein purity was further assessed by fast protein liquid chromatography, followed by SDS- PAGE and Coomassie blue staining.

For GST pull-down assays, cell lysate/His-tagged protein and GST fusion proteins were incubated together with glutathione-agarose beads. Complexes recovered from the beads were resolved by SDS-PAGE and analyzed by western blotting.

## FM4-64 dye imaging

Hippocampal neurons grown on various substrates were starved in neurobasal medium for 30 min and transferred to extracellular solution (145 mM NaCl, 10 mM HEPES, 8 mM glucose, 3 mM $CaCl_2$, 2 mM $MgCl_2$ and 3 mM KCl) before loading with the fluorescent dye FM 4-64FX (Invitrogen, Carlsbad, CA). Time-lapse images of neurons loaded with 20 μM FM4-64 were acquired as 10 μm z-series stacks (spaced at 0.5 μm) at 2 min intervals for 40 min using Yokogawa CSU-X1 spinning-disk confocal microscopy (Zeiss). Image analysis of FM dye intensity of the region of interest in cell bodies was performed by NIH ImageJ software. All pictures were taken at equal exposure for control and experimental groups.

## Measurements of QD-BDNF internalization

Quantum dot-labeled BDNF (QD-BDNF) was prepared as previously described (*Xie et al., 2012*). Briefly, streptavidin-conjugated QD655 (Invitrogen, Carlsbad, CA) mixed with human BDNF-biotin (Alomone Labs, Jerusalem, Israel) at a molar ratio of 1:2 was incubated overnight at 4°C. Unbound BDNF was separated from QD-BDNF with Sephacryl S-300 HR beads (Sigma-Aldrich, Saint Louis, MO), and the elution fractions (in 20 mM HEPES buffer, pH 7.2) with the highest fluorescence were regarded as the purified QD-BDNF. HEK293T cells transfected with TrkB expression constructs or hippocampal neurons were exposed to 0.1 nM QD-BDNF in 2% BSA containing extracellular solution for 30 min, followed by a period of washout before image acquisition. The fluorescence images of internalized QD-BDNF were acquired by spinning-disk confocal microscopy (Zeiss, multiple emission set: E460 SPUVv2, EX; 475DCXRU dichroic mirror, BS; D655/40 m, EM) in the presence of the quencher QSY-21 (2 μM) to prevent signals of extracellular QD-BDNF. We quantified amounts of BDNF-Qdots taken up by cells and correlated them with TrkB expression in transfected cells. BDNF-Qdot uptake by HEK293T cells requires TrkB expression, but TrkB levels do not show a linear correlation with numbers of intracellular BDNF-Qdots. Based on TrkB levels seen in our experiments, the amount of receptor is already saturating and sufficient to accurately assess BDNF-Qdot uptake by HEK293T cells. We omitted analysis of cells whose TrkB intensity is relatively low (cut-off = 5% of

highest TrkB intensity in each group). This adjustment does not alter overall observation that loss of paxillin expression significantly reduces BDNF-Qdot uptake.

## cDNA microarray and data validations

Total RNA of cortical neurons growing on gels of differing stiffness and glass was extracted at different time-points using a QuickGene RNA cultured cell kit from Kurabo Industries Ltd. (Osaka, Tokyo). RNA concentrations were quantified using a NanoDrop Spectrophotometer ND-1000 (NanoDrop Technologies, Wilmington, ED). RNA quality was assessed using a 2100 Bioanalyzer (Agilent Technologies, Santa Clara, CA) based on the following criteria: RNA integrity (number) >8.0, rRNA ratio (28S/18S) >1.8, and the proportion of 28S and 18S (amount) >40%. cDNA probe labeling and hybridization on rat gene expression microarray slides (Agilent Technologies, Santa Clara, CA) was conducted according to the manufacturers' protocols and carried out in the Institute of Molecular Biology Genomics Core Facility. Microarray data (19956 entrez genes) were analyzed using Gene-Spring GX software (version 12.1, Agilent). In brief, gene expression levels were subjected to quantile normalization and averaged from three independent experiments. Gene modules of the differential expressions were identified using a volcano plot (filtering criteria: >1.5 fold change, $p < 0.05$, $t$ test), followed by Gene Ontology analysis (cut off: $p < 0.02$; *Appendix 1—table 1*). Validation of results was achieved by overlaying the microarray-identified genes with a global molecular network (the IPA knowledge base), which revealed that the 114 genes were eligible to generate networks with a p-value<0.02. We prioritized the 114 genes based on their significance (*p*-value), and 'endocytosis-related reaction' came out as the top-ranked GO term for molecular functions among genes in the list. Three out of 37 genes that were selectively upregulated in 5 hr 0.1 kPa cultures are directly linked to endocytic processes. Based on significant shared GO terms, the correlations between the stiffness of substrates and the expression levels of genes associated with endocytosis, adhesion, neuronal development and cytoskeleton (see *Appendix 1—table 2*) were further verified by using QuantiGene Plex 2.0 assays according to the manufacturer's instructions (Affymetrix, Santa Clara, CA). The two housekeeping genes, GAPDH and HPRT, were used for data normalization. GEO accession number of the microarray data set is GSE102350.

## In utero electroporation

In utero electroporation followed previously described procedures (*Saito and Nakatsuji, 2001*), with minor modifications. Timed-pregnant Sprague-Dawley rats were anesthetized at E17.5 with isoflurane, and the uterine horns were exposed by way of a laparotomy. Saline solution containing the expression plasmid of interest (2 mg/ml) together with the dye Fast Green (0.3 mg/ml; Sigma-Aldrich, Saint Louis, MO) was injected (1–2 μl) through the uterine wall into one of the lateral ventricles of the embryos. The embryo's head was electroporated by tweezer-type circular electrodes across the uterus wall, and five electrical pulses (50 V, 50 ms duration at 100 ms intervals) were delivered with a square-wave electroporation generator (model ECM 830, BTX Inc.). The uterine horns were then returned to the abdominal cavity, the wall and skin were sutured, and the embryos continued normal development. Control embryos were electroporated with the tdTomato construct together with the GFP construct (1:2 ratio), and experimental embryos were electroporated with DynII$^{K44A}$ or PXN-shRNA (both sequence C and D) (see also Supplemental Materials), control scramble-shRNA, and the shRNA-resistant construct (PXN-shRNA-R), each in addition to the tdTomato construct. Control and experimental E20 embryos were obtained from the same litter and the injections were always made into the left and right ventricles, respectively, for later identification. Maximum intensities of the z-projection of tile images (20 μm thickness) were attained from spinning-disk confocal microscopy (Zeiss). Animal protocols were approved by the Animal Care and Use Committee of Academia Sinica.

## Membrane floatation assay

Dissected embryonic rat brain and heart at E17.5 were sonicated (four cycles of 20 s on and 10 s off) and mixed with 90% (w/w) sucrose to a final concentration of 45% sucrose in detergent-free lysis buffer (20 mM Tris-HCl, 5 mM MgCl$_2$ and 1 mM DTT with Complete protease inhibitor). Samples (1.5 ml) were overlaid with 1.5 ml of 35% sucrose and 9 ml of 5% sucrose and centrifuged at 175,000

x *g* for 18 hr at 4°C. Fractions (300 μl each) were collected from the top and subjected to immuno-blot analysis.

## Statistical analysis

To choose the statistical test for comparisons between two datasets, we first examined whether the data in each set was normally distributed using a Jarque-Bera test. A *t* test was used for normally distributed datasets. One-way ANOVA followed by a Dunnett's post hoc *test* was used for comparisons involving multiple datasets.

## Acknowledgements

We thank Hwai-Jong Cheng for discussions and critical reading of the manuscript. We thank R Thakar, S Li, M Nasir, and D Liepmann (University of California, Berkeley, CA) for help with PDMS microfluidic molds for making patterned substrates. We thank the Research Center for Applied Sciences, Academia Sinica, for help on atomic force microscopy measurements. We thank the IBC Mass Spectrometry Facilities (Institute of Biological Chemistry, Academia Sinica) for help with LTQ-Orbitrap data acquisition and additional technical assistance. We thank the IMB Genomics Core Facilities (Institute of Molecular Biology, Academia Sinica) for microarray analysis. This work was supported in part by a grant from the Ministry of Science and Technology of Taiwan.

## Additional information

### Funding

| Funder | Grant reference number | Author |
|---|---|---|
| Ministry of Science and Technology, Taiwan | Research Grant | Pei-Lin Cheng |

The funders had no role in study design, data collection and interpretation, or the decision to submit the work for publication.

### Author contributions

Ting-Ya Chang, Writing—original draft, Designed and carried out PA gel preparation, AFM measurements, biochemistry experiments, plasmid constructions, in utero electroporation and collected the data, Assisted with preparing the manuscript; Chen Chen, Carried out biochemistry experiments and collected the data; Min Lee, Assisted with biochemistry experiments; Ya-Chu Chang, Carried out smFISH experiments and collected the data; Chi-Huan Lu, Assisted with smFISH experiments and data analysis; Shao-Tzu Lu, Assisted with hippocampal cultures; De-Yao Wang, Assisted with lamellipodium phenotyping and PA gel preparation and data analysis; Aijun Wang, Designed PA gel preparation, Assisted with AFM measurement; Chin-Lin Guo, Developed mathematical model, Assisted with preparing the manuscript; Pei-Lin Cheng, Investigation, Wrote the paper, Deigned all experiments, Carried out all aspects of experiments and collected the data, Supervised the project

### Author ORCIDs

Aijun Wang [iD] http://orcid.org/0000-0002-2985-3627
Pei-Lin Cheng [iD] http://orcid.org/0000-0002-0090-8153

### Decision letter and Author response

Decision letter https://doi.org/10.7554/eLife.31101.033
Author response https://doi.org/10.7554/eLife.31101.034

## Additional files

### Supplementary files

• Transparent reporting form
DOI: https://doi.org/10.7554/eLife.31101.022

## Major datasets

The following dataset was generated:

| Author(s) | Year | Dataset title | Dataset URL | Database, license, and accessibility information |
|---|---|---|---|---|
| Chang TY, Chen C, Chang YC, Lu CH, Lee M, Lu ST, Wang DY, Wang A, Guo CL, Cheng PL | 2017 | Substrate stiffness-dependent gene expression profile for cortical neurons | https://www.ncbi.nlm.nih.gov/geo/query/acc.cgi?acc=GSE102350 | Publicly available at the NCBI Gene Expression Omnibus (accession no: GSE10 2350) |

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

# Appendix 1

DOI: https://doi.org/10.7554/eLife.31101023

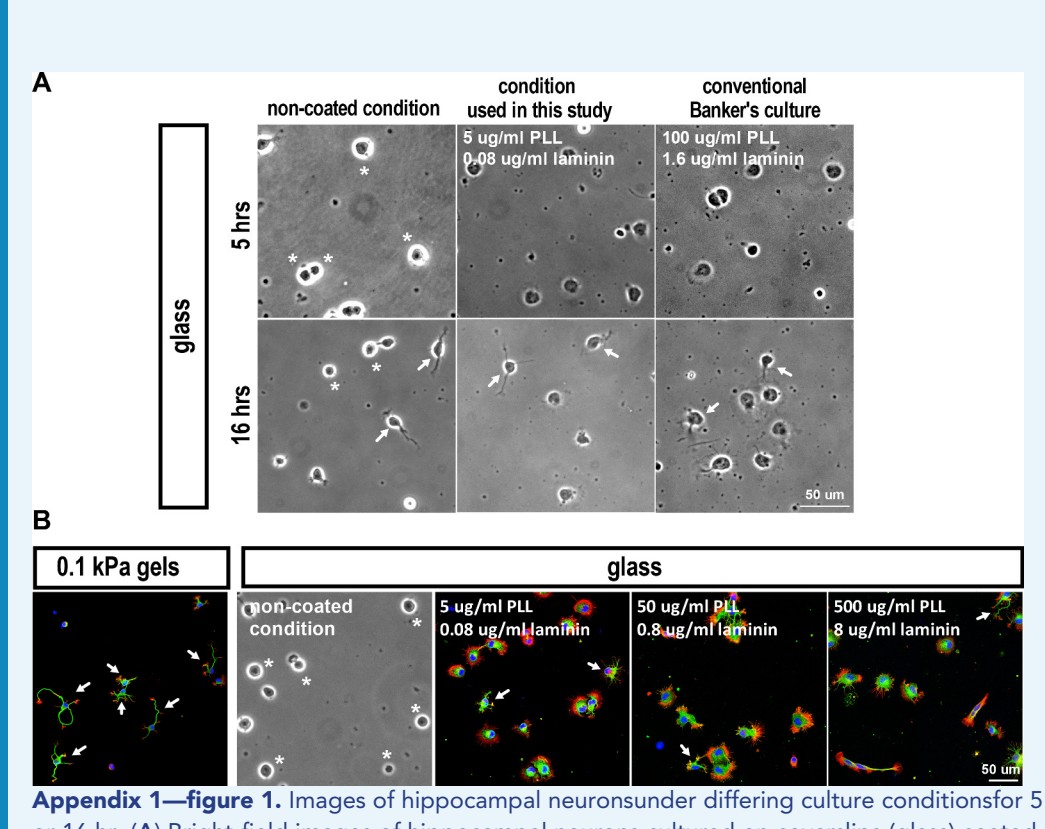

**Appendix 1—figure 1.** Images of hippocampal neuronsunder differing culture conditionsfor 5 or 16 hr. (**A**) Bright-field images of hippocampal neurons cultured on coverslips (glass) coated under differing conditions for 5 or 16 hr, as indicated. (**B**) Images of neurons grown on 0.1 kPa gels or coverslips with varied coatings for 5 hr, stained for F-actin (phalloidin; red), microtubules (anti-Tuj-1; green), and nuclei (DAPI; blue). Asterisks mark cells with no lamellipodial protrusions. Arrows mark neurite-bearing neurons.

DOI: https://doi.org/10.7554/eLife.31101.024

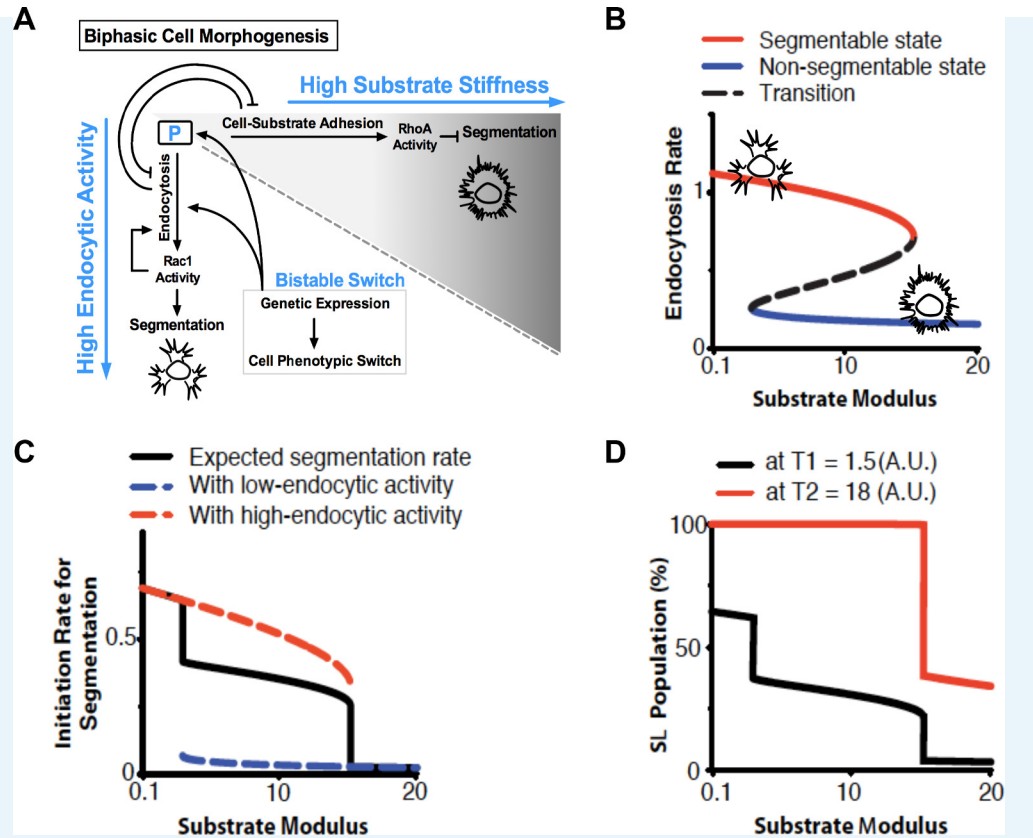

**Appendix 1—figure 2.** Proposed model for the biphasic behavior in neuronal morphogenesis (related to *Figure 3*). (**A**) The probability of segmentation depends on a 'P' factor that can bind to both endocytosis and adhesion complexes. When bound to the former, P factor enhances endocytosis and cells activate Rac1 and expression of factors required for the endocytic machinery (including P factor itself), providing a positive feedback that is modulated by substrate softness to accelerate segmentation. On stiff substrates, P factor is sequestered to cell-substrate adhesions, promoting a low-endocytosis and non-segmentable state. (**B**) Numerical simulations show that, based on the proposed model, cells can form a bistable switch between a high (Red line) and a low (Blue line) endocytosis state, dependent on the substrate modulus. (**C**) Numerical results show that the probability of segmentation depends on the substrate modulus and endocytic activity. (**D**) Numerical results show delayed onset of lamellipodium segmentation in cells grown on rigid substrates. For (**B–D**), see below model for details and parameters.

DOI: https://doi.org/10.7554/eLife.31101.025

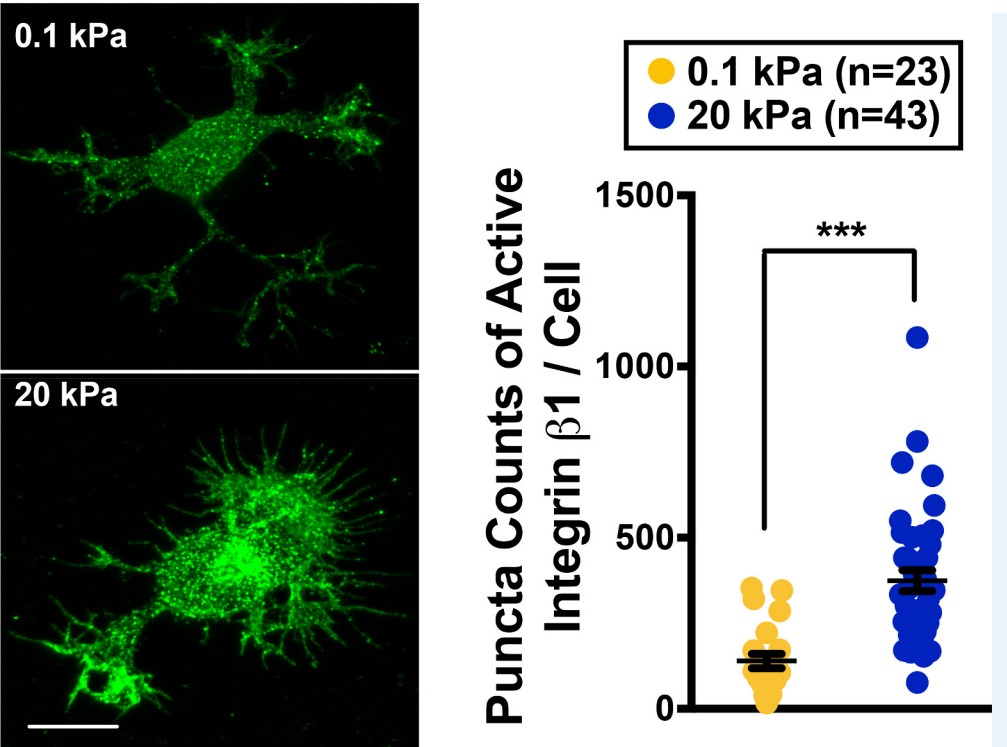

**Appendix 1—figure 3.** Active Integrin β1 staining of neurons grown on 0.1 kPa or 20 kPa gels. Right panel shows representative images of neuronal cultures stained with antibody specific against activated integrin β1 at the 16 hr time points after cell plating. Scale bar: 5 μm. Dot plot: quantification for puncta density of activated integrin 1 (n = 23–43 cells each, ***p<0.001; *t* test).

DOI: https://doi.org/10.7554/eLife.31101.026

## Basic assumptions of our model

To construct our model, we made the following assumptions: (a) There is a common factor (hereafter referred to as the 'P factor') that increases the probability of segmentation (and hence neurite formation); (b) Both adhesion (for cell substrates) and endocytosis complexes bind to the P factor; (c) Most P factors are sequestered to either the adhesion or the endocytosis complexes; (d) Association of P factor with the endocytosis complex enhances endocytosis, by which cells can up-regulate the activation of Rac1 and the expression of endocytosis-related molecules (including P factor); (e) Increasing substrate stiffness enhances cell-substrate adhesion, thereby sequestering P factor from binding to the endocytosis complex and down-regulating the endocytic rate; (f) Expression levels of P factor and the endocytic machinery are co-regulated; and (g) The probability of segmentation depends on Rac1 activity.

To simplify our analysis, we grouped the endocytic complex (or machinery) and the adhesion complex (or machinery) into single entities denoted as X and Y, respectively. Because we assumed that the expression of P factor and the endocytic machinery is co-regulated, we set the total expression level of P factor [$P_{Total}$] (represented by the concentration inside the cell) as being proportional to the total expression level of X [$X_{Total}$] according to $P_{Total} \approx e_{PX} \times X_{Total}$, where $e_{PX}$ is a constant.

The chemical balance between the association and dissociation of P factor with X and Y follows:

$$[P] + [X] \xrightarrow{K_{px}} [PX],$$
$$[PX] \xrightarrow{K_{px\_diss}} [P] + [X].$$

(1)

$$[P] + [Y] \xrightarrow{K_{py}} [PY],$$
$$[PY] \xrightarrow{K_{py\_diss}} [P] + [Y].$$

(2)

with the constraint $[PX] + [PY] + [P] = [P_{Total}]$, $[PX] + [X] = [X_{Total}]$, and $[PY] + [Y] = [Y_{Total}]$, where $[P_{Total}]$, $[X_{Total}]$, and $[Y_{Total}]$ are the total expression levels of P factor, the adhesion complex machinery, and the endocytosis complex machinery, respectively. We assumed that most P factors are sequestered to either the endocytosis or the adhesion complex, and free-form P factor is limited (i.e., $[P] \ll 1$). As such, the quasi-steady state concentration of $PX$ and $PY$ can be estimated as:

$$[PX] = \frac{k_{px}[P][X_{Total}]}{k_{px\_diss} + k_{px}[P]} \approx k_{px}k_{px\_diss}^{-1}[P][X_{Total}],$$

$$[PY] = \frac{k_{py}[P][Y_{Total}]}{k_{py\_diss} + k_{py}[P]} \approx k_{py}k_{py\_diss}^{-1}[P][Y_{Total}]$$

(3)

Consequently, we have:

$$[P] \approx \frac{[P_{Total}]}{1 + k_{py}k_{py\_diss}^{-1}[Y_{Total}] + k_{px}k_{px\_diss}^{-1}[X_{Total}]},$$

$$\rightarrow [PX] \approx \frac{k_{px}k_{px\_diss}^{-1}e_{PX}[X_{Total}]^2}{1 + k_{py}k_{py\_diss}^{-1}[Y_{Total}] + k_{px}k_{px\_diss}^{-1}[X_{Total}]}.$$

(4)

For simplicity, we further assumed that the endocytic rate is proportional to $[PX]$.

$$Endocytic\ rate \ \propto [PX]$$

(5)

## Analysis of two-state behavior

Next, we considered the equation for endocytosis-induced up-regulation of the expression of P factor and the endocytic machinery. We used a simple approach in that the endocytosis-induced up-regulation follows Michaelis–Menten kinetics (**Michaelis et al., 2011**). Based on the assumption that the endocytic rate is proportional to $[PX]$ (**Equation (5)**), we have:

$$\frac{d[X_{Total}]}{dt} = k_{\deg}\left(X_0 + X_{up}\frac{[PX]}{PX_T + [PX]} - [X_{Total}]\right),$$

(6)

where $k_{\deg}$ is the degradation rate, $X_0$ is the baseline expression in the absence of endocytosis-induced up-regulation, $X_{up}$ is the efficacy for endocytosis-induced up-regulation. Michaelis–Menten kinetics use a constant $PX_T$ as the threshold for $[PX]$ to reach half the maximum level of up-regulation. Substituting **Equation (4)** into **Equation (6)**, we have:

$$\frac{d[X_{Total}]}{k \cdot dt} = X_0 + \frac{X_{up}k_{up}k_{px\_diss}^{-1}e_{PX}[X_{Total}]^2}{PX_T\left(1 + k_{py}k_{py_{diss}}^{-1}[Y_{Total}] + k_{px}k_{px\_diss}^{-1}[X_{Total}]\right) + k_{px}k_{px\_diss}e_{PX}[X_{Total}]^2} - [X_{Total}].$$

(7)

We further simplified **Equation (7)** by setting $t = k_{\deg}t$, $\beta = e_{PX}X_0/PX_T$, $\alpha = e_{PX}X_{up}/PX_T$, $x = e_{PX}[X_{Total}]/PX_T$, and $y = e_{PX}(1 + k_{py}[Y_{Total}]/k_{py\_diss})/(k_{px}PX_T/k_{px\_diss})$ and obtained:

$$\frac{dx}{d\tau} = \beta + \frac{\alpha x^2}{y + x + x^2} - x,$$

(8)

$$Endocytosis\ rate\ \propto [PX] \approx PX_T \frac{x^2}{x+y} \propto \frac{x^2}{x+y}. \tag{9}$$

Here, variable $x$ is related to the expression level of the endocytic machinery (including P factor). Variable $y$ is related to cell-substrate adhesion and increases with substrate stiffness. For simplicity, we approximated $y$ with a linear dependence on the substrate modulus; namely, $y = y_0 + y_1 \times$ substrate modulus, where $y_0$ and $y_1$ are constants. The steady-state solutions of **Equation (8)** can be found by solving its null-clines in the $x$-$y$ plane, which read:

$$y = \frac{\alpha x^2}{x - \beta} - x(x+1). \tag{10}$$

An example of **Equation (10)** is illustrated in **Appendix 1—figure 2B–D**, in which we use a linear relationship to link the substrate modulus with the variable $y$ (i.e., $y = y_0 + y_1 \times$ substrate modulus) and to set the endocytic rate as a function of $x$ and $y$ (**Equation (9)**). Analysis of **Equation (10)** revealed that for a given $y$, two stable steady-state solutions of $x$ could coexist (the two solid lines in Figure S6B) with one unstable transition-state solution (the dotted line in Figure S6B), provided that the following criterion is satisfied:

$$(\alpha - 2\beta - 1)^3 \geq 27\alpha\beta^2 \tag{11}$$

Under such a condition, cells possess the ability to perform a bistable switch between a high-$x$ state and a low-$x$ state for a given $y$ (i.e., substrate stiffness). Here, the high-$x$ state corresponds to the endocytosis-dominant state where the cells exhibit a higher endocytic rate, higher Rac1 activation, and higher expression of the endocytic machinery (including P factor). In comparison, the low-$x$ state corresponds to the adhesion-dominant state with lower Rac1 activation and lower expression of the endocytic machinery. For convenience, hereafter we define $S(x, y)$ as the state for a given set of $(x, y)$. Furthermore, for any given $y$, we define $S_{end}(x_{end}(y), y)$ as the corresponding endocytosis-dominant state, with $x_{end}(y)$ as the steady-state solution from **Equation (10)**. Likewise, we define $S_{adh}(x_{adh}(y), y)$ as the adhesion-dominant state for a given $y$, with $x_{adh}(y)$ as the steady-state solution. Similarly, $S_{tx}(x_{tx}(y), y)$ defines the transition state for a given $y$, with $x_{tx}(y)$ as the steady-state solution.

### Statistics of two-state behavior

The distribution of the proposed two states at a given substrate modulus (i.e., $y$) can be estimated by rewriting **Equation (8)** into a Langevin equation followed by adopting a free energy approach (**Lemons and Gythiel, 1997**; **Mossa and Clementi, 2007**). The 'free energy' is then used to estimate the probability of finding one cell at a given state. Furthermore, the free energy can be used to estimate the rate at which the cells switch from one state to another. Specifically, we approximated **Equation (8)** as the derivative of a free energy, $F(x, y)$, and added a time-dependent Gaussian white noise $\eta$ of width $D$ to the system:

$$\frac{dx}{d\tau} = \beta + \frac{\alpha x^2}{y + x + x^2} - x + \eta(\tau) = -\frac{dF(x,y)}{dx} + \eta(\tau), \tag{12}$$

where

$$\langle \eta(\tau)\eta(\tau')\rangle = D\delta(\tau - \tau')$$
$$\langle \eta(\tau)\rangle = 0 \tag{13}$$

$$F(x,y) = C - (\alpha + \beta)x - \frac{x^2}{2} + \frac{\alpha}{2}\mathrm{In}(y + x + x^2) + \frac{\alpha}{2} \times$$

$$\begin{cases} \dfrac{(y-1/2)}{\sqrt{|y-1/4|}}\mathrm{In}\left(\dfrac{x+1/2 - \sqrt{|y-1/4|}}{x+1/2 + \sqrt{|y-1/4|}}\right) & y < 1/4 \\[3mm] \dfrac{1}{x+1/2} & y = 1/4, \\[3mm] \dfrac{(y-1/2)}{\sqrt{|y-1/4|}}\tan^{-1}\left(\dfrac{x+1/2}{|y-1/4|}\right) & y > 1/4 \end{cases} \tag{14}$$

with $C$ as an integral constant. The probability of finding one cell at a state $S(x, y)$, denoted as $f(x, y)$, is then specified by the free energy as:

$$f(x,y) \propto \exp[F(x,y)/D], \tag{15}$$

or

$$f(x,y) = \frac{e^{-F(x,y)/D}}{\int dx e^{-F(x,y)/D}}. \tag{16}$$

Note that in Eqn. [16], $x$ does not need to be the steady-state solution from **Equation (10)**. For the bistable regime where cells are allowed to select one of the steady states at a given $y$ (provided that **Equation (11)** is satisfied) or switch from one state to another, the rate at which the switch occurs can be estimated by Transition State Theory, Eyring theory, Arrhenius' law, or Kramers' law (**Goychuk et al., 2010**; **Wodkiewicz, 1984**). For instance, for a cell in the endocytosis-dominant state, the average rate at which it switches to the adhesion-dominant state, $k_{end \to adh}$, can be calculated by the mean first passage time by which the cell moves from the endocytosis-dominant state $S_{end}(x_{end}(y), y)$ to the transition state $S_{tx}(x_{tx}(y), y)$, $\tau_{end \to tx}$, in the free energy landscape $F(x, y)$. Likewise, the average rate at which the cell switches from the adhesion-dominant state to the endocytosis-dominant state, $k_{adh \to end}$, can be calculated by the mean first passage time $\tau\alpha_{dh \to tx}$. Following Transition State Theory, Eyring theory, Arrhenius' law, or Kramers' law (**Goychuk et al., 2010**; Wodkiewicz, 1984), we establish the switch rate at a given $y$ as:

$$\begin{aligned} k_{end \to adh}(y) &\sim k_0 \times \exp[(F(x_{end}(y),y) - F(x_{tx}(y),y))/D], \\ k_{adh \to end}(y) &\sim k_0 \times \exp[(F(x_{adh}(y),y) - F(x_{tx}(y),y))/D] \end{aligned} \tag{17}$$

where $k_0$ is a constant.

## Segmentation rate in the two-state system

In our model, we assumed that the probability of segmentation depends on Rac1 activity, which is primarily induced by endocytosis. To estimate the segmentation rate, hereafter defined as $k_s(x, y)$, we approximated it as being proportional to the endocytic rate **Equations (5 and 9)**, namely:

$$Segmentation\ rate\ k_s(x,y) \propto Endocytosis\ rate \sim \frac{x^2}{x+y}. \tag{18}$$

Examples of the segmentation rates from **Equation (18)** are illustrated in Figure S6C, where the rates for the endocytosis-dominant and adhesion-dominant states are plotted. For a given $y$, the expected rate of segmentation in a population of cells was estimated as:

$$\text{Expected Segmentation rate } \langle k_s(y) \rangle \quad \sim \quad \frac{f(x_{adh}(y))\frac{x_{adh}^2}{x_{adh}+y} + f(x_{end}(y))\frac{x_{end}^2}{x_{end}+y}}{f(x_{adh}(y),y) + f(x_{end}(y),y)}$$

$$= \frac{e^{-F(x_{adh}(y),y)/D}\frac{x_{adh}^2}{x_{adh}+y} + e^{-F(x_{end}(y),y)/D}\frac{x_{end}^2}{x_{end}+y}}{e^{-F(x_{adh}(y),y)/D} + e^{-F(x_{end}(y),y)/D}}. \tag{19}$$

## Segmentation dynamics in the two-state system

Finally, we estimated the dynamics of a population of cells that progressively switched to the segmented state. To do so, we assumed that cells were initially in the adhesion-dominant state, and defined the density of cells in the adhesion-dominant state as $n_{adh}(\tau)$ for a given time $\tau$, the density of cells in the endocytosis-dominant state as $n_{end}(\tau)$, and the density of cells that formed segments as $n_{seg}(\tau)$, with the following constraints: $n_{adh}(0) = 1$, $n_{end}(0) = 0$, and $n_{adh}(\tau)+n_{end}(\tau)+n_{seg}(\tau)=1$. The dynamics for these cells are as follows:

$$\frac{dn_{adh}}{d\tau} = -k_s(x_{adh}(y),y)n_{adh} - k_{adh\to end}(y)n_{adh} + k_{end\to adh}(y)n_{end},$$

$$\frac{dn_{end}}{d\tau} = -k_s(x_{end}(y),y)n_{end} + k_{adh\to end}(y)n_{adh} + k_{end\to adh}(y)n_{end}, \tag{20}$$

$$\frac{dn_{seg}}{d\tau} = -k_s(x_{adh}(y),y)n_{adh} + k_s(x_{end}(y),y)n_{end}.$$

Examples of the segmentation dynamics in a population of cells are illustrated in Figure S6D, where the contributions from cells in the endocytosis-dominant or the adhesion-dominant states is plotted separately.

## Parameters used in the figures

To simulate the segmentation dynamics illustrated in *Appendix 1—figure 2*, we set $\alpha = 2.5$, $\beta = 0.1$, $k_0 = 1$, $D = 1$, and $y = 0.7 + 10^{-5} \times$ substrate rigidity, and used *Equations (10, 17, 18 and 20)*.

**Appendix 1—table 1.** Gene ontology analysis.

| | | 5 hr | | | 16 hr |
|---|---|---|---|---|---|
| | | *Receptor-mediated endocytosis* | | | *Cell differentiation* |
| | ** | receptor-mediated endocytosis | | * | forebrain neuron fate commitment |
| | * | endocytic adaptor activity | | | *Cytoskeleton* |
| | * | clathrin adaptor activity | | ** | intermediate filament binding |
| | ** | regulation of receptor recycling | | * | protein localization to cytoskeleton |
| 0.1 kPa | *** | positive regulation of receptor recycling | | * | protein localization to microtubule cytoskeleton |
| | | *Vesicle transport* | | | *Cell aggregation* |
| | ** | endosomal vesicle fusion | | * | cell aggregation |
| | * | vesicle-mediated transport | | | GTPase signaling pathway |
| | *** | response to lipid | | ** | G-protein coupled neurotensin receptor activity |

*Appendix 1—table 1 continued on next page*

*Appendix 1—table 1 continued*

| | | 5 hr | 16 hr |
|---|---|---|---|
| **0.1 kPa** | | *External stimulus* | |
| | *** | response to external stimulus | |
| | ** | regulation of cell-substrate adhesion | |
| | * | laminin-1 binding | |
| | | *Neuronal development* | |
| | * | axon midline choice point recognition | |
| | ** | regulation of long term synaptic depression | |
| | | *Cell polarity* | |
| | * | maintenance of cell polarity | |
| | * | positive regulation of Wnt signaling pathway, planar cell polarity pathway | |
| **0.1 kPa** | | *Cell migration* | |
| | ** | regulation of cell migration | |
| | * | positive regulation of cell migration | |
| | ** | regulation of cell motility | |
| | * | positive regulation of cell motility | |
| | | *GTPase signaling pathway* | |
| | ** | regulation of Rho-dependent protein serine/threonine kinase activity | |
| | * | adenylate cyclase-activating G-protein coupled receptor signalling pathway | |
| | | *Cytoskeleton* | |
| | * | neurofilament cytoskeleton | |
| **20 kPa** | | *Nervous system development* | *Nervous system development* |
| | * | central nervous system development | * central nervous system neuron development |
| | * | regulation of dendrite development | ** negative regulation of neurogenesis |
| | ** | regulation of neurogenesis | * neuronal action potential propagation |
| | ** | regulation of nervous system development | ** regulation of neuron projection development |
| | * | pyramidal neuron development | * axon midline choice point recognition |
| | * | cell projection | * axon guidance receptor activity |
| | ** | internode region of axon | *Signaling transduction* |
| | ** | cellular developmental process | * protein kinase A signaling |
| | * | regulation of cell development | ** positive regulation of phosphatidylinositol 3-kinase signalling |

*Appendix 1—table 1 continued on next page*

*Appendix 1—table 1 continued*

| | 5 hr | | 16 hr |
|---|---|---|---|
| **20 kPa** | *Cell differentiation* | *** | second-messenger-mediated signaling |
| | * cell differentiation | | *Cell differentiation* |
| | * pyramidal neuron differentiation | ** | neuron fate commitment |
| | * regulation of cell morphogenesis involved in differentiation | ** | central nervous system neuron differentiation |
| | *Vesicle transport* | ** | negative regulation of neuron differentiation |
| | ** late endosome | | *Morphogenesis* |
| | * regulation of lipid transport | * | cell morphogenesis |
| | *Cell migration* | * | positive regulation of dendritic spine morphogenesis |
| **20 kPa** | ** locomotory behavior | | *Cell adhesion* |
| | * negative regulation of neuron migration | * | response to mechanical stimulus |
| | *GTPase signaling pathway* | ** | extracellular matrix binding |
| | * Rho guanyl-nucleotide exchange factor activity | * | positive regulation of cell-substrate adhesion |
| | ** Rho protein signal transduction | ** | laminin-1 binding |
| | * positive regulation of adenylate cyclase activity involved in G-protein coupled receptor signalling pathway | **** | positive regulation of cell adhesion |
| | *Cytoskeleton* | ** | protein binding involved in cell adhesion |
| | * regulation of actin cytoskeleton organization | | *Cell migration* |
| **20 kPa** | * regulation of actin filament-based process | ** | positive regulation of cell migration |
| | | * | negative regulation of cell adhesion involved in substrate-bound cell migration |
| | | ** | positive regulation of locomotion |
| | | | *Neuronal regeneration* |
| | | * | negative regulation of axon regeneration |
| | | * | negative regulation of neuron projection regeneration |
| | | | Transport |
| | | ** | regulation of transport |
| | | | *Cytoskeleton* |
| **20 kPa** | | * | actin filament binding |
| | | ** | actin filament bundle assembly |
| | *p<0.05; **p<0.01; ***p<0.001; ****p<0.0001. t test | * | ruffle assembly |

DOI: https://doi.org/10.7554/eLife.31101.027

**Appendix 1—table 2.** Microarray analysis and identification of substrate modulus-regulated genes in hippocampal neurons.

| Gene symbol | Accession number | Gene name | 0.1 kPa | 1 kPa | 20 kPa | Glass |
|---|---|---|---|---|---|---|

*Appendix 1—table 2 continued on next page*

Appendix 1—table 2 continued

| | | Gene symbol | Accession number | Gene name | 0.1 kPa | 1 kPa | 20 kPa | Glass |
|---|---|---|---|---|---|---|---|---|
| 5 hr | | inhba | NM_017128 | inhibin beta-A | 12.45 | 11.81 | 11.83 | 11.65 |
| | • | cdc42bpa | NM_053657 | CDC42 binding protein kinase alpha | 11.45 | 10.91 | 10.78 | 11.22 |
| | | ubr2 | NM_001178071 | ubiquitin protein ligase E3 component n-recognin 2 | 10.82 | 10.38 | 10.21 | 10.20 |
| | | mx1 | NM_173096 | myxovirus (influenza virus) resistance 1 | 10.72 | 10.06 | 9.91 | 9.85 |
| | • | myo6 | XM_006226497 | myosin VI (probe 1) | 10.58 | 10.03 | 9.96 | 10.40 |
| | | tagln2 | NM_001013127 | transgelin 2 | 10.31 | 9.84 | 9.72 | 10.67 |
| | | gbp2 | NM_133624 | guanylate binding protein 2, interferon-inducible | 10.17 | 9.54 | 8.86 | 9.37 |
| | | col1a2 | NM_053356 | collagen, type I, alpha 2 | 10.14 | 9.53 | 9.45 | 10.45 |
| 5 hr | | cxcl16 | NM_001017478 | chemokine (C-X-C motif) ligand 16 | 9.89 | 9.34 | 8.78 | 9.63 |
| | | ifi27l2b | NM_206846 | interferon, alpha-inducible protein 27 like 2B | 9.88 | 8.80 | 7.95 | 9.11 |
| | | bpifb1 | NM_001077680 | BPI fold containing family B, member 1 | 9.84 | 8.73 | 8.44 | 7.44 |
| | • | pik3r1 | NM_013005 | phosphoinositide-3-kinase, regulatory subunit 1 (alpha) | 9.82 | 9.28 | 9.16 | 9.36 |
| | | sorcs3 | NM_001106367 | sortilin-related VPS10 domain containing receptor 3 | 9.76 | 8.86 | 8.74 | 9.57 |
| | | slc13a3 | NM_022866 | solute carrier family 13 (sodium-dependent dicarboxylate transporter), member 3 | 9.55 | 9.14 | 8.94 | 8.41 |
| | | igf2 | NM_031511 | insulin-like growth factor 2 | 9.41 | 8.58 | 8.59 | 8.99 |
| | | mbp | NM_001025291 | myelin basic protein, transcript variant 1 | 9.40 | 8.33 | 8.59 | 8.31 |
| 5 hr | • | plvap | NM_020086 | plasmalemma vesicle associated protein | 9.32 | 8.43 | 7.98 | 9.33 |
| | | rsad2 | NM_138881 | radical S-adenosyl methionine domain containing 2 | 9.30 | 8.82 | 8.11 | 9.19 |
| | • | dab2 | NM_024159 | disabled 2, mitogen-responsive phosphoprotein | 8.94 | 8.46 | 7.97 | 8.14 |
| | | isg15 | NM_001106700 | ISG15 ubiquitin-like modifier | 8.83 | 8.17 | 7.24 | 7.91 |
| | • | myo6 | XM_006226497 | myosin VI (probe 2) | 8.65 | 8.10 | 8.00 | 8.42 |
| | • | robo3 | NM_001108135 | roundabout, axon guidance receptor, homolog 3 (Drosophila) | 8.49 | 7.81 | 7.56 | 6.89 |
| | | samd9l | XM_001069386 | sterile alpha motif domain containing 9-like | 8.48 | 7.84 | 7.63 | 9.07 |
| | | nid1 | XM_213954 | nidogen 1 | 8.10 | 7.02 | 7.06 | 7.65 |

Appendix 1—table 2 continued

| | | Gene symbol | Accession number | Gene name | 0.1 kPa | 1 kPa | 20 kPa | Glass |
|---|---|---|---|---|---|---|---|---|
| 5 hr | □ | nexn | NM_139230 | nexilin (F actin binding protein) | 7.91 | 6.72 | 6.90 | 7.15 |
| | | calcr | NM_053816 | calcitonin receptor | 7.90 | 6.30 | 5.73 | 7.13 |
| | | irg1 | NM_001107282 | immunoresponsive gene 1 | 7.88 | 7.40 | 6.54 | 6.79 |
| | | lgals3bp | NM_139096 | lectin, galactoside-binding, soluble, 3 binding protein | 7.86 | 7.51 | 7.14 | 7.08 |
| | | ccl6 | NM_001004202 | chemokine (C-C motif) ligand 6 | 7.79 | 7.03 | 6.29 | 5.86 |
| | | aebp1 | NM_001100970 | AE binding protein 1 | 7.69 | 6.99 | 6.93 | 7.55 |
| | | tp53inp1 | NM_181084 | tumor protein p53 inducible nuclear protein 1 | 7.63 | 7.32 | 7.01 | 7.55 |
| | | trpc3 | NM_021771 | transient receptor potential cation channel, subfamily C, member 3 | 7.36 | 6.90 | 6.76 | 5.94 |
| 5 hr | | oasl2 | NM_001009682 | 2'-5' oligoadenylate synthetase-like 2 | 7.33 | 6.69 | 6.20 | 6.39 |
| | | nqo1 | NM_017000 | NAD(P)H dehydrogenase, quinone 1 | 7.33 | 6.87 | 6.55 | 6.70 |
| | | dst | XM_006226778 | dystonin | 7.22 | 6.71 | 6.50 | 6.70 |
| | • | arap1 | XM_006223427 | ArfGAP with RhoGAP domain, ankyrin repeat and PH domain 1 | 6.97 | 6.45 | 6.30 | 6.55 |
| | | gcgr | NM_172091 | glucagon receptor | 6.94 | 6.49 | 5.55 | 5.99 |
| 5 hr | • | arhgef19 | NM_001108692 | Rho guanine nucleotide exchange factor (GEF) 19 | 8.96 | 9.30 | 9.59 | 8.92 |
| | • | map2k6 | NM_053703 | mitogen-activated protein kinase kinase 6 | 8.75 | 9.22 | 9.37 | 8.83 |
| | • | gpr56 | NM_152242 | G protein-coupled receptor 56 | 8.72 | 9.00 | 9.40 | 8.67 |
| | | tst | NM_012808 | thiosulfate sulfurtransferase | 8.66 | 9.26 | 9.29 | 9.12 |
| | | six5 | XM_003753221 | SIX homeobox 5 | 7.96 | 8.63 | 8.64 | 8.21 |
| | | mbp | NM_001025289 | myelin basic protein, transcript variant 6 | 7.89 | 8.46 | 8.58 | 8.18 |
| | | atp7a | NM_052803 | ATPase, Cu++ transporting, alpha polypeptide | 7.66 | 8.29 | 8.85 | 7.95 |
| | | cyb561a3 | NM_001014164 | cytochrome b561 family, member A3 | 6.97 | 7.53 | 7.65 | 7.23 |
| 5 hr | | dnah8 | XM_001079004 | dynein, axonemal, heavy chain 8 | 6.71 | 7.52 | 7.43 | 7.01 |
| | | scrt2 | NM_001191905 | scratch family zinc finger 2 | 6.04 | 6.36 | 6.64 | 6.18 |
| | | crhr1 | NM_030999 | corticotropin releasing hormone receptor 1 | 5.95 | 6.42 | 6.60 | 5.91 |
| | | bmx | NM_001109016 | BMX non-receptor tyrosine kinase | 5.95 | 6.54 | 6.56 | 6.77 |
| | ○ | celsr1 | XM_006242168 | cadherin, EGF LAG seven-pass G-type receptor 1 | 5.88 | 6.47 | 6.60 | 6.60 |
| | | timp3 | NM_012886 | TIMP metallopeptidase inhibitor 3 | 5.77 | 6.03 | 6.47 | 6.89 |
| | • | prex1 | NM_001135718 | phosphatidylinositol-3,4,5-trisphosphate-dependent Rac exchange factor 1 | 5.48 | 5.89 | 6.07 | 5.64 |
| | | clcn2 | NM_017137 | chloride channel, voltage-sensitive 2 | 5.38 | 5.84 | 6.09 | 5.39 |

Appendix 1—table 2 continued on next page

*Appendix 1—table 2 continued*

| | | Gene symbol | Accession number | Gene name | 0.1 kPa | 1 kPa | 20 kPa | Glass |
|---|---|---|---|---|---|---|---|---|
| 5 hr | • | crebl2 | NM_001015027 | cAMP responsive element binding protein-like 2 | 5.15 | 5.45 | 5.79 | 5.25 |
| | | neurog3 | NM_021700 | neurogenin 3 | 4.80 | 5.18 | 5.43 | 5.57 |
| | | fam161a | NM_001013876 | family with sequence similarity 161, member A | 4.80 | 5.34 | 5.60 | 5.29 |
| 16 hr | | snx10 | NM_001013085 | sorting nexin 10 | 9.69 | 9.49 | 8.86 | 9.23 |
| | | col2a1 | NM_012929 | collagen, type II, alpha 1 | 9.69 | 9.23 | 8.91 | 9.29 |
| | | ftcd | NM_053567 | formimidoyltransferase cyclodeaminase | 9.67 | 9.13 | 8.68 | 9.28 |
| | | tfap2c | NM_201420 | transcription factor AP-2 gamma | 8.27 | 8.14 | 7.67 | 8.29 |
| | | ntsr2 | NM_022695 | neurotensin receptor 2 | 7.65 | 7.25 | 6.98 | 7.37 |
| | ○ | itgb4 | NM_013180 | integrin, beta 4 | 7.34 | 6.98 | 6.75 | 6.68 |
| 16 hr | | caprin1 | NM_001012185 | cell cycle associated protein 1 | 11.46 | 11.53 | 12.08 | 11.56 |
| | | inhba | NM_017128 | inhibin beta-A | 11.31 | 11.91 | 12.45 | 11.98 |
| | • | robo2 | NM_032106 | roundabout, axon guidance receptor, homolog 2 (Drosophila) | 11.03 | 11.29 | 11.69 | 11.32 |
| | | mkks | NM_001008353 | McKusick-Kaufman syndrome | 10.70 | 11.05 | 11.29 | 11.30 |
| | | itpr1 | NM_001007235 | inositol 1,4,5-trisphosphate receptor, type 1 | 10.32 | 10.80 | 11.22 | 10.83 |
| | | igf1 | NM_001082479 | insulin-like growth factor 1 | 10.21 | 10.61 | 11.53 | 10.35 |
| | | aif1 | NM_017196 | allograft inflammatory factor 1 | 9.82 | 10.34 | 10.52 | 10.52 |
| | | gucy1a3 | XM_006232518 | guanylate cyclase 1, soluble, alpha 3 | 9.75 | 9.91 | 10.42 | 9.96 |
| | | cnksr2 | NM_021686 | connector enhancer of kinase suppressor of Ras 2 | 9.73 | 9.88 | 10.32 | 9.73 |
| 16 hr | | ptpn6 | NM_053908 | protein tyrosine phosphatase, non-receptor type 6 | 9.67 | 10.15 | 10.32 | 10.18 |
| | | lcp1 | NM_001012044 | lymphocyte cytosolic protein 1 | 9.13 | 9.64 | 10.14 | 9.39 |
| | | olig2 | NM_001100557 | oligodendrocyte lineage transcription factor 2 | 9.10 | 9.56 | 10.61 | 9.83 |
| | | jag1 | NM_019147 | jagged 1 | 9.05 | 9.21 | 9.69 | 9.27 |
| | | fst | NM_012561 | follistatin | 8.99 | 9.65 | 10.05 | 9.36 |
| | | gucy1b3 | NM_012769 | guanylate cyclase 1, soluble, beta 3 | 8.98 | 9.08 | 9.69 | 9.30 |
| | | igf1 | NM_001082479 | insulin-like growth factor 1 | 8.86 | 9.28 | 10.15 | 9.08 |
| | | olig1 | NM_021770 | oligodendrocyte transcription factor 1 | 8.85 | 9.32 | 9.81 | 9.82 |
| | | npy1r | NM_001113357 | neuropeptide Y receptor Y1 | 8.61 | 9.28 | 9.76 | 9.28 |
| 16 hr | | tnf | NM_012675 | tumor necrosis factor | 8.57 | 8.95 | 9.49 | 8.71 |
| | ○ | cdh11 | NM_053392 | cadherin 11 | 8.33 | 8.59 | 9.06 | 8.64 |
| | ○ | nrcam | NM_013150 | neuronal cell adhesion molecule | 8.28 | 8.92 | 8.91 | 8.78 |
| | | spp1 | NM_012881 | secreted phosphoprotein 1 | 8.20 | 9.52 | 11.57 | 8.78 |
| | | ccl4 | NM_053858 | chemokine (C-C motif) ligand 4 | 8.16 | 8.94 | 9.37 | 8.52 |
| | | nid1 | XM_213954 | nidogen 1 | 7.85 | 7.97 | 8.86 | 9.12 |
| | | cd53 | NM_012523 | Cd53 molecule | 7.83 | 8.44 | 9.00 | 8.13 |
| | • | ntf3 | NM_031073 | neurotrophin 3 | 7.78 | 7.87 | 8.45 | 7.96 |
| | | il18 | NM_019165 | interleukin 18 | 7.28 | 8.07 | 8.42 | 8.09 |

*Appendix 1—table 2 continued on next page*

*Appendix 1—table 2 continued*

| | | Gene symbol | Accession number | Gene name | 0.1 kPa | 1 kPa | 20 kPa | Glass |
|---|---|---|---|---|---|---|---|---|
| 16 hr | | fcgr3a | NM_207603 | Fc fragment of IgG, low affinity IIIa, receptor | 7.23 | 7.91 | 8.71 | 7.77 |
| | • | cdk6 | NM_001191861 | cyclin-dependent kinase 6 | 6.81 | 7.11 | 7.73 | 7.57 |
| | ○ | itgb2 | NM_001037780 | integrin, beta 2 | 6.63 | 7.09 | 7.61 | 6.91 |
| | | ly86 | NM_001106128 | lymphocyte antigen 86(probe1) | 6.45 | 7.29 | 7.53 | 7.22 |
| | | ly96 | NM_001024279 | lymphocyte antigen 96 | 6.02 | 6.39 | 6.70 | 6.48 |
| | | ccr5 | NM_053960 | chemokine (C-C motif) receptor 5 | 5.96 | 6.28 | 6.72 | 5.83 |
| | | ptprc | NM_138507 | protein tyrosine phosphatase, receptor type, C | 5.95 | 6.82 | 7.46 | 6.54 |
| | | ly86 | NM_001106128 | lymphocyte antigen 86(probe 2) | 5.94 | 6.79 | 7.00 | 6.86 |
| | | loc360228 | NM_001003706 | WDNM1 homolog | 4.40 | 6.20 | 6.02 | 5.80 |

○ Adhesion

• Endocytosis

• Neuritogenesis-related signaling

□ Cytoskeleton.

DOI: https://doi.org/10.7554/eLife.31101.028

**Appendix 1—table 3.** Paxillin-associating protein identified by liquid chromatography-tandem mass spectrometry.

| Category | Protein ID | Score | MW | MWobs |
|---|---|---|---|---|
| Endocytosis | clathrin heavy chain 1 | 68 | 190 | 190 |
| | dynamin-1-like protein isoform X17 | 84 | 80 | 80 |
| Adhesion | talin-1 | 159 | 270 | 270 |
| Cytoskeleton | drebrin isoform X3 | 90 | 72 | 120 |
| dynamics | alpha-internexin | 70 | 56 | 60 |
| | citron Rho-interacting kinase | 63 | 235 | 270 |
| | protein unc-45 homolog B isoform X1 | 243 | 103 | 105 |
| Transcription/ | matrin-3 | 152 | 94 | 120 |
| Translation | elongation factor 1-alpha 1 | 94 | 50 | 49 |
| | eukaryotic translation initiation factor 3 subunit A | 114 | 163 | 180 |
| | eukaryotic translation initiation factor 3 subunit B | 57 | 90 | 120 |
| | eukaryotic translation initiation factor 3, subunit 6 interacting protein | 51 | 66 | 65 |
| | heterogeneous nuclear ribonucleoprotein M, isoform X2 | 98 | 64 | 75 |
| | AT-rich interactive domain-containing protein 1B isoform X3, partial | 58 | 58 | 63 |
| | heterogeneous nuclear ribonucleoprotein A2/B1 | 181 | 38 | 37 |
| Protein quality | heat shock protein 60 kDa precrusor | 171 | 61 | 60 |
| control | heat shock cognate 71 kDa protein | 182 | 71 | 75 |
| | heat shock protein 90 kDa alpha, class B member 1 | 222 | 83 | 95 |
| | T-complex protein 1 subunit alpha | 79 | 60 | 60 |
| | T-complex protein 1 subunit gamma | 79 | 60 | 60 |
| | protein unc-45 homolog B isoform X1 | 243 | 103 | 105 |
| | stress-70 protein | 70 | 74 | 75 |

*Appendix 1—table 3 continued*

| Category | Protein ID | Score | MW | MWobs |
|---|---|---|---|---|
| Signaling transduction | calcium/calmodulin-dependent protein kinase II, beta 3 isoform | 67 | 65 | 65 |
| DNA repair | fanconi anemia group C protien homolog | 73 | 64 | 60 |
| Metabolism | glyceraldehyde-3-phosphate dehydrogenase | 192 | 36 | 36 |

DOI: https://doi.org/10.7554/eLife.31101.029

