## [Decision Letter]

Thank you for submitting your article "Paxillin facilitates timely neurite initiation on soft-substrate environments by interacting with the endocytic machinery" for consideration by *eLife*. Your article has been favorably evaluated by Marianne Bronner (Senior Editor) and three reviewers, one of whom, Kang Shen, is a member of our Board of Reviewing Editors. The following individual involved in review of your submission has agreed to reveal her identity: Bianxiao Cui (Reviewer #2).

The reviewers have discussed the reviews with one another and the Reviewing Editor has drafted this decision to help you prepare a revised submission.

Summary:

The manuscript by Chang et al. investigated an interesting observation that embryonic neurons cultured on the soft substrate grow out neurites much earlier. Through pharmacological inhibitor treatments, genetic manipulations (in cells and in animals), and biochemical probes, the authors established a strong correlation between early neurite outgrowth and the endocytosis process. In particular, the authors identified a crucial protein player paxillin that mediates the correlation. Paxillin is widely known for its role in focal adhesion. Using co-IP and other methods, this paper demonstrated that paxillin also participates in clathrin-mediated endocytosis by interacting with CIP4, a previously unknown role for paxillin. Their evidence suggests that two processes (focal adhesion and endocytosis) compete for paxillin, depending on the substrate rigidity. These discoveries led to an interesting hypothesis that Paxillin acts as a switch between adhesion and endocytosis. The manuscript also contains a large of amount of well-carried-out work, and the evidence is very convincing. Overall, this manuscript is of high quality and the reviewers support the potential publication of this manuscript in *eLife* given that the authors can address the issues raised below. Most of the issues can be dealt with by rewriting. However, appropriate experimental data can also be added if they help to clarify the issues.

Essential revisions:

1) We are not completely convinced that paxillin functions as a "molecular switch" – the authors do not perform experiments where they manipulate paxillin's binding state to endocytic or adhesion molecules and observe an ability to switch between neurite outgrowth/no outgrowth. Indeed, overexpression of CIP4 did not affect neural migration (Figure 5—figure supplement 2) although the polarity of neurons was not quantified here. As part of the argument for the "molecular switch" model, the authors suggest that paxillin binds preferentially with endocytic factors when neurons are grown on soft cultures to initiate neurite outgrowth, whereas it binds preferentially with adhesion factors when neurons are grown on stiff cultures. This conclusion is drawn from colocalization in immunostaining in Figure 3, as well as co-IP in Figure 4. However, the expression levels of endocytic and adhesion factors also change depending on the stiffness of the substrate – could the increased localization of paxillin with vinculin and p-FAK on stiff substrates just reflect an increase in expression?

2) Furthermore, is there evidence that such a substrate-based switch between neurite outgrowth and non-growth is physiologically relevant? Are there tissues in the developing brain or embryo with high enough stiffness to trigger the adhesion "switch" in paxillin (2016 AFM showed 400 Pa in retinotectal system)?

3) The evidence for vinculin vs. CIP4 competitive binding to paxillin could use more validation – it appears that vinculin is capable of binding to the LD1 domain, and the authors find it necessary to use a highly truncated paxillin to disrupt binding of both vinculin and CIP4.

Minor points [abridged]:

[...] 8) Since this manuscript suggests that paxillin facilitates neurite growth by interacting with CIP4 and participating in endocytosis, it implies that CIP4 also has a positive role in neurite growth. However, a previous published paper (cited below) suggests that CIP4 inhibits neurite outgrowth, a conclusion that is opposite of this manuscript. All work done in the previous was performed on glass substrates. Is it possible that CIP4 also functions differently in substrates of different softness? Can the authors reconcile their conclusion with the previous study? "The F-BAR Protein CIP4 Inhibits Neurite Formation by Producing Lamellipodial Protrusions"Current Biology 22, 494-501, March 20, 2012.

---

## [Author Response]

Essential revisions:1) We are not completely convinced that paxillin functions as a "molecular switch" – the authors do not perform experiments where they manipulate paxillin's binding state to endocytic or adhesion molecules and observe an ability to switch between neurite outgrowth/no outgrowth. Indeed, overexpression of CIP4 did not affect neural migration (Figure 5—figure supplement 2) although the polarity of neurons was not quantified here. As part of the argument for the "molecular switch" model, the authors suggest that paxillin binds preferentially with endocytic factors when neurons are grown on soft cultures to initiate neurite outgrowth, whereas it binds preferentially with adhesion factors when neurons are grown on stiff cultures. This conclusion is drawn from colocalization in immunostaining in Figure 3, as well as co-IP in Figure 4. However, the expression levels of endocytic and adhesion factors also change depending on the stiffness of the substrate – could the increased localization of paxillin with vinculin and p-FAK on stiff substrates just reflect an increase in expression?

A) In response to this comment, we have added genetic experiments that employ additional paxillin variants, namely, paxillin-∆LD1 and paxillin-∆LIM3-4, to manipulate paxillin binding affinity and assess effects on formation of proper neurite precursors. Based on domain mapping results shown in Figure 4 and revised Figure 4—figure supplement 1, we conclude that the paxillin-∆LD1 variant favors CIP4/clathrin association and that paxillin-∆LIM3-4 favors vinculin association. Using this approach, we found that ectopic expression of paxillin-∆LIM3-4 in 0.1 kPa cultures increased the neuritogenesis-delaying BL phenotype, whereas paxillin-∆LD1 expression reduced the size of the BL population in 20 kPa cultures. In contrast, ectopic expression of the paxillin LD 1-3 variant, which cannot bind vinculin or CIP4, had no significant effect on neurite outgrowth. These results demonstrate that manipulating paxillin’s binding affinity to favor either endocytic or adhesion molecules can change the potential of newborn neurons to form proper neurites. These new data are now shown in the new Figure 7—figure supplement 1.

B) We agree that greatly increased levels of adhesion factors may contribute to sequestering paxillin over less abundant endocytic factors on a stiff substrate. To minimize ambiguity, we now comment on this issue in the Discussion in reporting colocalization experiments (Please see Discussion, second paragraph). Overall, our findings support the idea that gene expression patterns and binding of accessory proteins, such as paxillin, that favor either endocytotic or adhesion reactions govern morphogenetic states of newly differentiated neurons following cell-matrix contact (for a schematic showing proposed model, please see Figure 7).

3) We thank the referee for thoughtful suggestions. We have now added quantitative measurements of polarity of in utero-electroporated (IUE) neurons to revised Figure 5—figure supplement 2. Briefly, we categorized IUE neurons into three groups based on the number and morphology of neuronal processes: 1) polarized neurons – cells bearing observable tailing processes (axons) and leading processes (dendrites); 2) unpolarized neurons – cells bearing short neurites but lacking polarity; and 3) no-neurite neurons – cells with no neurites. We observed no significant difference in the number of cells in each group between control and GFP-CIP4 overexpressing neurons. These new data are shown in the revised Figure 5—figure supplement 2.

2) Furthermore, is there evidence that such a substrate-based switch between neurite outgrowth and non-growth is physiologically relevant? Are there tissues in the developing brain or embryo with high enough stiffness to trigger the adhesion "switch" in paxillin (2016 AFM showed 400 Pa in retinotectal system)?

We thank the referees for this insightful question. We have now added comments relevant to this topic to the Discussion (please see Discussion, fourth paragraph). Based on studies reporting the natural stiffness of mouse embryonic cerebral cortex at E16.5 to E18.5 (Iwashita, M., et al., Development 2014), we surmise that a neuronal switch favoring paxillin/endocytic factor binding is required and activated immediately after embryonic neuronal differentiation to ensure timely neurite initiation. This switch may also promote departure of cells from neurogenic regions, such as the ventricular and subventricular zones, whose stiffness is relatively high compared to the cortical plate at E18.5. However, at a later developmental time-point (since brain stiffness increases with age) and at synaptic contact sites or neuromuscular junctions, the paxillin adhesion switch could take effect and regulate formation of contacts with postsynaptic cells or decrease the capacity for neurite outgrowth. For example, synapses at neuro-muscular junctions where repetitive muscle contraction can endow much greater localized stiffness than that seen in brain cortex may favor paxillin/adhesion machinery association. A paxillin-mediated switch from neuritogenesis to the development of cell adhesion could stabilize synapses and connectivity between pre-synaptic and post-synaptic cells.

3) The evidence for vinculin vs. CIP4 competitive binding to paxillin could use more validation – it appears that vinculin is capable of binding to the LD1 domain, and the authors find it necessary to use a highly truncated paxillin to disrupt binding of both vinculin and CIP4.

A) We agree that the competition model needs support. Given that CIP4 forms a dimer in vitro and in vivo (as verified by small angle X-ray scattering analysis), we feel that increased CIP4 binding to paxillin LIM domains may spatially hinder vinculin from accessing the LD motifs. We investigated this possibility using two new sets of experiments. First, we conducted additional competition assays using paxillin variants, i.e., full-length paxillin, ∆LD1 paxillin, and ∆LIM3-4 domain. Overall, these studies showed that the presence of the vinculin-binding LD1 domain does not alter competition by CIP4 or vinculin for paxillin binding. In contrast, the CIP4-binding LIM 3-4 region is required for CIP4 competitive capacity, as CIP4 could not out-compete vinculin for binding to the paxillin-∆LIM3-4 construct. This new data is shown in Figure 4—figure supplement 1. Secondly, we undertook additional domain-mapping analyses using a series of N-terminal-truncated paxillin constructs (shown in Figure 4 and Figure 4—figure supplement 1). These analyses showed that vinculin and CIP4 primarily associate with LD motifs (LD1, 2, and/or 4; Turner et al., J. Cell Biol., 1999) and the LIM domain, respectively. Nevertheless, we also noted that deletion of the paxillin LIM3-4 domain reduced its affinity for vinculin (see comparison of full-length paxillin versus paxillin-∆LIM3-4 constructs in Figure 4), suggesting that the paxillin LIM3-4 domain is required for strong binding of vinculin to LD motifs. On the other hand, CIP4 shows detectable binding to the LD motif on the LD1-5 construct. These findings potentially explain why the CIP4-paxillin interaction interferes with the vinculin-LD association. Based on a previous report (Turner et al., J. Cell Biol., 1999) and our new results, we have revised the schematic model (shown in Figure 4) and parts of the Results (subsection “Endocytic factor CIP4 and adhesion factor vinculin are potential competitors for paxillin binding”, last paragraph) to avoid misinterpretation of the competitive-binding mechanism.

B) We agree that vinculin and CIP4 interaction domains do not fully overlap on paxillin. Although the LIM-domain-truncated paxillin LD1-5 construct contains only vinculin-binding LD domains, it retains residual binding to CIP4. Therefore, we used the paxillin LD1-3 construct, which contains no LIM domain and lacks the last two LD domains, as the negative control. Also, we currently do not know if different forms of regulation, such as post-translational modification, govern paxillin binding preferences to CIP4 or vinculin in newborn neurons grown on different substrates.

Minor points [abridged]:[...] 8) Since this manuscript suggests that paxillin facilitates neurite growth by interacting with CIP4 and participating in endocytosis, it implies that CIP4 also has a positive role in neurite growth. However, a previous published paper (cited below) suggests that CIP4 inhibits neurite outgrowth, a conclusion that is opposite of this manuscript. All work done in the previous was performed on glass substrates. Is it possible that CIP4 also functions differently in substrates of different softness? Can the authors reconcile their conclusion with the previous study? "The F-BAR Protein CIP4 Inhibits Neurite Formation by Producing Lamellipodial Protrusions"Current Biology 22, 494-501, March 20, 2012.

We appreciate the referee pointing out that CIP4 may function differently on substrates of differing softness. We observed altered CIP4 distribution along the enlarged lamellar edge (as reported in Saengsawang et al., 2012), but only when CIP4-GFP-expressing neurons were cultured on 20 kPa gels or coverslips (states unfavorable for neurite initiation (see **Figure 5-figure supplement 2B**). This altered CIP4 distribution may simply reflect CIP4's ability to bind lipid bi-layers where its accumulation may mark or reflect membrane curvature. In addition, we observed interactions of paxillin with CIP4 and endocytic machineries in neurons growing on a soft environment but not on coverslips, implying their function and associating complexes depend on cell content/state. Therefore, we feel that previous studies complement rather than challenge our findings. CIP4 over-expression also did not inhibit neurite formation in cortical progenitor cells or in neurons grown on soft substrates (0.1 kPa gels), conditions in which paxillin co-localizes with the endocytic machinery (please see **Figures 3D, 3E, and Figure 5-figure supplement 2C**). Based on these results, we are confident that paxillin and CIP4 function in the endocytic machinery in newborn neurons grown in or on soft tissue environments.